# AAV ablates neurogenesis in the adult murine hippocampus

Stephen Johnston[1,2†], Sarah L Parylak[2†], Stacy Kim[2,3†], Nolan Mac[4], Christina Lim[2], Iryna Gallina[2], Cooper Bloyd[2], Alexander Newberry[5], Christian D Saavedra[2], Ondrej Novak[6], J Tiago Gonçalves[7,8], Fred H Gage[2]*, Matthew Shtrahman[3]*

[1]Neurosciences Graduate Program, University of California, San Diego, La Jolla, United States; [2]Laboratory of Genetics, Salk Institute for Biological Studies, La Jolla, United States; [3]Department of Neurosciences, University of California, San Diego, La Jolla, United States; [4]Department of Biology, University of California, San Diego, La Jolla, United States; [5]Department of Physics, University of California, San Diego, La Jolla, United States; [6]Laboratory of Experimental Epileptology, Department of Physiology, Second Faculty of Medicine, Charles University, Prague, United Kingdom; [7]Ruth L. and David S. Gottesman Institute for Stem Cell Biology and Regenerative Medicine, Albert Einstein College of Medicine, Bronx, United States; [8]Dominick P. Purpura Department of Neuroscience, Albert Einstein College of Medicine, Bronx, United States

*For correspondence:
gage@salk.edu (FHG);
mshtrahman@ucsd.edu (MS)

[†]These authors contributed equally to this work

Competing interests: The authors declare that no competing interests exist.

**Abstract** Recombinant adeno-associated virus (rAAV) has been widely used as a viral vector across mammalian biology and has been shown to be safe and effective in human gene therapy. We demonstrate that neural progenitor cells (NPCs) and immature dentate granule cells (DGCs) within the adult murine hippocampus are particularly sensitive to rAAV-induced cell death. Cell loss is dose dependent and nearly complete at experimentally relevant viral titers. rAAV-induced cell death is rapid and persistent, with loss of BrdU-labeled cells within 18 hr post-injection and no evidence of recovery of adult neurogenesis at 3 months post-injection. The remaining mature DGCs appear hyperactive 4 weeks post-injection based on immediate early gene expression, consistent with previous studies investigating the effects of attenuating adult neurogenesis. In vitro application of AAV or electroporation of AAV2 inverted terminal repeats (ITRs) is sufficient to induce cell death. Efficient transduction of the dentategyrus (DG)– without ablating adult neurogenesis– can be achieved by injection of rAAV2-retro serotyped virus into CA3. rAAV2-retro results in efficient retrograde labeling of mature DGCs and permits in vivo two-photon calcium imaging of dentate activity while leaving adult neurogenesis intact. These findings expand on recent reports implicating rAAV-linked toxicity in stem cells and other cell types and suggest that future work using rAAV as an experimental tool in the DG and as a gene therapy for diseases of the central nervous system should be carefully evaluated.

## Introduction

The subgranular zone (SGZ) of the hippocampal dentate gyrus (DG) is one of only a few regions of the mammalian brain that continues to exhibit neurogenesis into adulthood. Adult-born dentate granule cells (abDGCs) are continuously generated from a pool of largely quiescent neural stem cells that undergo proliferation, differentiation, and fate specification before maturing into neurons that are indistinguishable from developmentally derived dentate granule cells (DGCs) (*Gonçalves et al., 2016*; *Kempermann et al., 2015*). These stem cells and their immature progeny are sensitive to environmental stimuli; their proliferation, development, and survival are regulated by multiple

intrinsic and extrinsic factors, including experience, stress, inflammation, and pharmacologic agents (see Materials and methods, *Gonçalves et al., 2016*; *Kempermann et al., 2015*; *Monje et al., 2003*; *Snyder et al., 2009*; *Vivar et al., 2016*). Numerous studies demonstrate that abDGCs are critical for maintaining the physiological activity of mature DGCs and contribute to hippocampus-dependent behaviors (*Clelland et al., 2009*; *Deng et al., 2009*; *Deng et al., 2010*; *Ikrar et al., 2013*; *Lacefield et al., 2012*; *Nakashiba et al., 2012*; *Sahay et al., 2011*; *Saxe et al., 2007*; *Tronel et al., 2012*). While the specific role that immature DGCs play in hippocampal function, including the formation of memories, is not fully established, progress has been achieved through recent work focused on precisely modulating and measuring the activity of immature and mature DGCs within the DG of animals during behavior (*Anacker et al., 2018*; *Danielson et al., 2016*; *Danielson et al., 2017*; *Hainmueller and Bartos, 2018*; *Hayashi et al., 2017*; *Kirschen et al., 2017*; *Leutgeb et al., 2007*; *Pilz et al., 2016*; *Senzai and Buzsáki, 2017*).

A key tool enabling many of these and other advances in in vivo neurophysiology is recombinant adeno-associated virus (rAAV). Wild-type AAV is a non-enveloped, single-stranded DNA virus endemic to humans and primates and has been previously proposed to have no known pathogenicity. This replication-defective virus contains a 4.7 kb genome that includes the Rep and Cap genes and a pair of palindromic 145 bp inverted terminal repeats (ITRs). The Rep and Cap genes can be supplied in trans to create space for incorporating transgenes of interest, yielding the widely used rAAV, which retains only the ITRs from the original wild-type genome. In experimental neuroscience, rAAV is often used to deliver a variety of genetically encoded tools, including actuators and sensors of neuronal function, to specific cell types and brain regions. In addition, rAAV's minimal viral genome and limited immunogenicity and toxicity have made it the vector of choice for human gene therapy (*Büning and Schmidt, 2015*; *Choudhury et al., 2017*; *Hocquemiller et al., 2016*; *Hudry and Vandenberghe, 2019*), including two FDA-approved therapies for disorders of the CNS (*Hoy, 2019*; *Smalley, 2017*).

Despite its safety profile, rAAV has increasingly been reported to demonstrate toxicity in some cell types (*Bockstael et al., 2012*; *Hinderer et al., 2018*; *Hirsch et al., 2011*; *Hordeaux et al., 2018*). However, the toxic effects of rAAV on abDGCs have previously not been assessed. Motivated by our own efforts to study the role of adult neurogenesis and the DG in learning and memory, we discovered that neural progenitor cells (NPCs) and immature neurons in the DG are highly susceptible to rAAV-induced death at a range of experimentally relevant titers (3 ᴇ11 gc/mL and above). This process appears to be cell autonomous and mediated by AAV2 ITRs, which are used nearly universally in rAAVs. Consistent with previous ablation studies, elimination of 4-week-old abDGCs by rAAV alters the activity of mature DGCs, resulting in DG hyperactivity (as indicated by cFOS expression). To circumvent this problem, we used the rAAV2-retro serotype (*Tervo et al., 2016*) to label DGCs in a retrograde manner, which avoids infection of susceptible cells and preserves adult neurogenesis. We demonstrate the utility of this delivery method by measuring the activity of mature DGCs in vivo using two-photon calcium imaging.

## Results

### rAAV eliminates abDGCs in a dose-dependent manner

In preliminary studies, we found that the delivery of calcium indicators via commonly used rAAV serotypes at doses equivalent to or below previously reported doses resulted in a dramatic qualitative loss of the immature neuron marker doublecortin (DCX) 2 weeks after viral injection (*Figure 1—figure supplement 1A*). This effect occurred following the injection of a variety of rAAV preparations, regardless of vector production facility (Salk Institute Viral Vector Core, University of Pennsylvania Vector Core, Addgene), purification method (iodixanol, CsCl), capsid serotype (AAV1 and AAV8; all incorporating AAV2 ITRs), promoter (CAG, Syn, CaMKIIa), and protein expression (GFP, jRGECO1a, a red calcium indicator, and mCherry) at doses typically required for the functional manipulation or visualization of DGCs in vivo. To systematically quantify the effect of rAAV transduction on abDGCs, we chose to inject a widely available, minimally expressing cre-recombinase-dependent virus (AAV1-CAG-flex-eGFP, U. Penn. and Addgene #51502) in non-cre-expressing wild-type C57BL/6J mice to mitigate any contributions from toxicity that might be attributed to protein expression. Mice received daily intraperitoneal injections of 5-bromo-2′-deoxyuridine (BrdU) for 3

days to label dividing cells; then 1 µL of 3E12 gc/mL rAAV was injected unilaterally into the DG immediately (0 day), or 1, 2, or 8 weeks later (schematic in *Figure 1A*). Cell survival on the virus-treated side depended on the age of BrdU-labeled cells when the virus was delivered ($F_{treatment \times time}(3,27)=29.0$, $p<0.001$; *Figure 1B*, last four columns). Cells that were 2 days old and younger at the time of injection were almost completely eliminated within 48 hr ($-83.9 \pm 6.7\%$, $p<0.001$, all

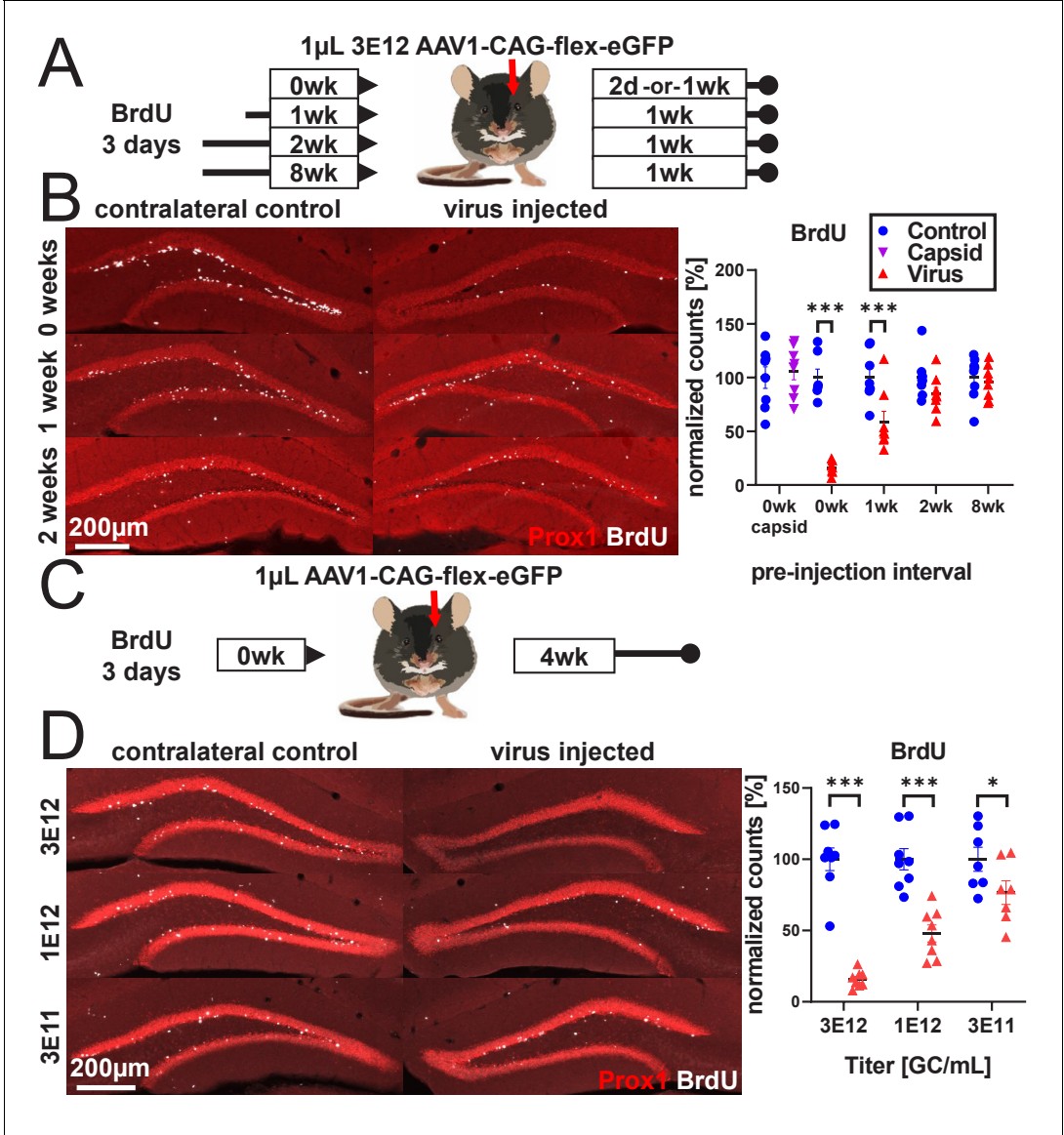

**Figure 1.** rAAV eliminates abDGCs in a dose-dependent manner. (A) Experimental design of rAAV injection into DG following indelible labeling of adult-born DGCs with BrdU. (B, left) Representative images showing Prox1 and BrdU used for quantification. (B, right) abDGCs birth-dated with BrdU for 3 days immediately preceding viral injection show near-complete elimination following rAAV injection; cells born 1 week before viral injection are reduced by ~50%. Cells born 2 weeks prior or more, or injection of empty AAV viral capsid, demonstrate no reduction. (C) Experimental design of dose-dependent attenuation of abDGCs by rAAV. (D, left) Representative images showing Prox1 and BrdU used for quantification. (D, right) A near complete ablation of BrdU+ cells is seen in the DG injected with 1 µL 3 E12 GC/mL rAAV, partial ablation of BrdU+ cells results from the injection of 1 µL 1 E12 GC/mL rAAV, and a small but significant reduction of adult neurogenesis results from injection of 1 µL 3 E11 GC/mL rAAV. All data are presented as mean ± s.e.m, significance reported as: *p<0.05, ***p<0.001.

The online version of this article includes the following source data and figure supplement(s) for figure 1:

**Source data 1.** Source data for *Figure 1*.

**Figure supplement 1.** rAAV-dependent toxicity.

**Figure supplement 1—source data 1.** Source data for *Figure 1—figure supplement 1*.

textual results reported as change relative to the mean of non-injected contralateral DG ± standard error of the mean difference, unless stated otherwise). Cells that were 7–9 days old were partially protected ($-41.3 \pm 6.3\%$, $p<0.001$), whereas cells that were 14–16 days old were largely protected with variable but non-significant loss ($-15.4 \pm 6.3\%$, n.s.). Fully mature abDGCs, approximately 8 weeks old (*Laplagne et al., 2007*; *Piatti et al., 2013*), also did not demonstrate significant loss at 1 week following AAV injection ($-4.5 \pm 6.3\%$, n.s.; *Figure 1B* last column). In the same tissue, Tbr2+ intermediate progenitors were lost even when mature BrdU+ cells were spared, demonstrating that the virus did not lack toxicity in these animals ($F_{treatment \times time}(3,27)=3.0$, $p<0.05$; 0 weeks: $-80.2 \pm 9.1\%$, $p<0.001$; 1 weeks: $-76.9 \pm 8.5\%$, $p<0.001$; 2 weeks: $-82.6 \pm 8.5\%$, $p<0.001$; 8 weeks: $-50.8 \pm 8.5\%$, $p<0.001$; *Figure 1—figure supplement 1B*). To determine the effect of viral attachment and penetration in rAAV-induced toxicity, we injected 1 µL of high titer (3.7 E13 capsids/mL) empty AAV viral capsid (University of North Carolina Viral Vector Core and Salk Institute Viral Vector Core) containing no viral DNA into the DG (*Figure 1B*, first column, *Figure 1—figure supplement 1C*). At 1 week post-injection, there was no effect on BrdU+ cells (2 days old at the time of empty capsid injection) relative to the contralateral control DG, although there was a mild decrease 4 weeks post-injection that did not recapitulate the severe loss observed with intact virus ($F_{treatment \times time}(1,14)=6.5$, $p<0.05$, 1 week post-injection: $6.2 \pm 5.7\%$, n.s., *Figure 1B*, first column; 4 weeks: $-14.5 \pm 5.7\%$, $p<0.05$, *Figure 1—figure supplement 1C*).

We then assessed the effect of titer on rAAV-induced cell loss. We labeled abDGCs for 3 days with BrdU and injected 1 µL of either 3 E12 gc/mL, 1 E12 gc/mL, or 3 E11 gc/mL rAAV on the final day of BrdU labeling (schematic in *Figure 1C*). Cell loss increased with increasing titer of virus injected ($F_{treatment \times titer}(2,20)=19.2$, $p<0.001$). A nearly complete ablation of BrdU+ cells was seen in the DG injected with 3 E12 gc/mL rAAV ($-84.3 \pm 6.7\%$, $p<0.001$), whereas partial ablation of BrdU+ cells resulted from the injection of 1 E12 gc/mL rAAV ($-52.1 \pm 6.7\%$, $p<0.001$), and a small but significant reduction of adult neurogenesis resulted from injection of 3E11 gc/mL rAAV ($-23.4 \pm 7.2\%$, $p<0.01$; *Figure 1D*). This dose-dependent pattern was matched by reductions in immature neuron marker DCX expression (*Figure 1—figure supplement 1D*).

## Developmental stage determines susceptibility to rAAV-induced cell loss

After determining the differential response of abDGCs to rAAV based on post-mitotic age, we determined which population of NPCs was susceptible to rAAV-induced loss. To accomplish this, we varied the post-injection interval and measured canonical early (Sox2), middle (Tbr2), and late (DCX) histological markers associated with abDGC development. Mice were unilaterally injected with 1 µL of 3 E12 gc/mL rAAV and sacrificed at 2 days, 1 week, or 4 weeks post-injection (schematic in *Figure 2A*). The number of Sox2+ cells within the SGZ was modestly decreased ($F_{treatment}(1,19)=15.5$, $p<0.001$; $F_{time}(2,19)=1.6$, n.s.; $F_{treatment \times time}(2,19)=2.7$, n.s.; *Figure 2B,C*). In contrast, Tbr2+ intermediate progenitor cells were almost entirely lost and did not show signs of recovery by 4 weeks post-injection ($F_{treatment}(1,19)=129.2$, $p<0.001$; $F_{time}(2,19)=0.1$, n.s.; $F_{treatment \times time}(2,19)=0.2$, n.s.; *Figure 2B,D*). Expression of the late premitotic and immature neuronal marker DCX showed progressive decline until near complete loss at 4 weeks post-injection ($F_{treatment \times time}(3,27)=12.8$, $p<0.001$; 2 days: $-27.7 \pm 7.5\%$, $p<0.01$; 1 week: $-58.7 \pm 7.0\%$, $p<0.001$; 4 weeks: $-92.0 \pm 7.5\%$, $p<0.001$; *Figure 2B,G*) and did not show signs of recovery 3 months post-injection ($-68.7 \pm 6.6\%$, $p<0.001$; *Figure 2G*, last column).

Significant loss of Tbr2+ and DCX+ cells occurred even in the most conservative experimental conditions, in which BrdU was absent (to prevent any synergistic toxicity between rAAV and BrdU) and when saline was injected contralaterally (to mimic any physical disruption due to the injection process itself) (*Figure 2—figure supplement 1A–E*). Injection of empty AAV viral capsid (*Figure 2E*) alone did not result in a reduction of Tbr2+ or DCX+ cells at 1 or 4 weeks post-injection (Tbr2: $F_{treatment}(1,10)=6.4E-5$, n.s. $F_{time}(1,10)=0.3$, n.s.; $F_{treatment \times time}(1,10)=1.8$, n.s; *Figure 2F*; DCX: $F_{treatment}(1,14)=3.9$, n.s.; $F_{time}(1,14)=0.7$, n.s.; $F_{treatment \times time}(1,14)=1.3$, n.s.; *Figure 2—figure supplement 1G*). These results demonstrate that viral DNA is required for cell loss.

To explore whether differences in progenitor subtype survival could be explained by differences in tropism or infectivity, we injected rAAV expressing GFP with the same capsid serotype (AAV1-CAG-GFP Addgene #:37825-AAV1) into dorsal DG and measured the fraction of DCX+, Tbr2+, and Sox2+ cells in the SGZ that express GFP 1 week after viral injection (*Figure 2—figure supplement*

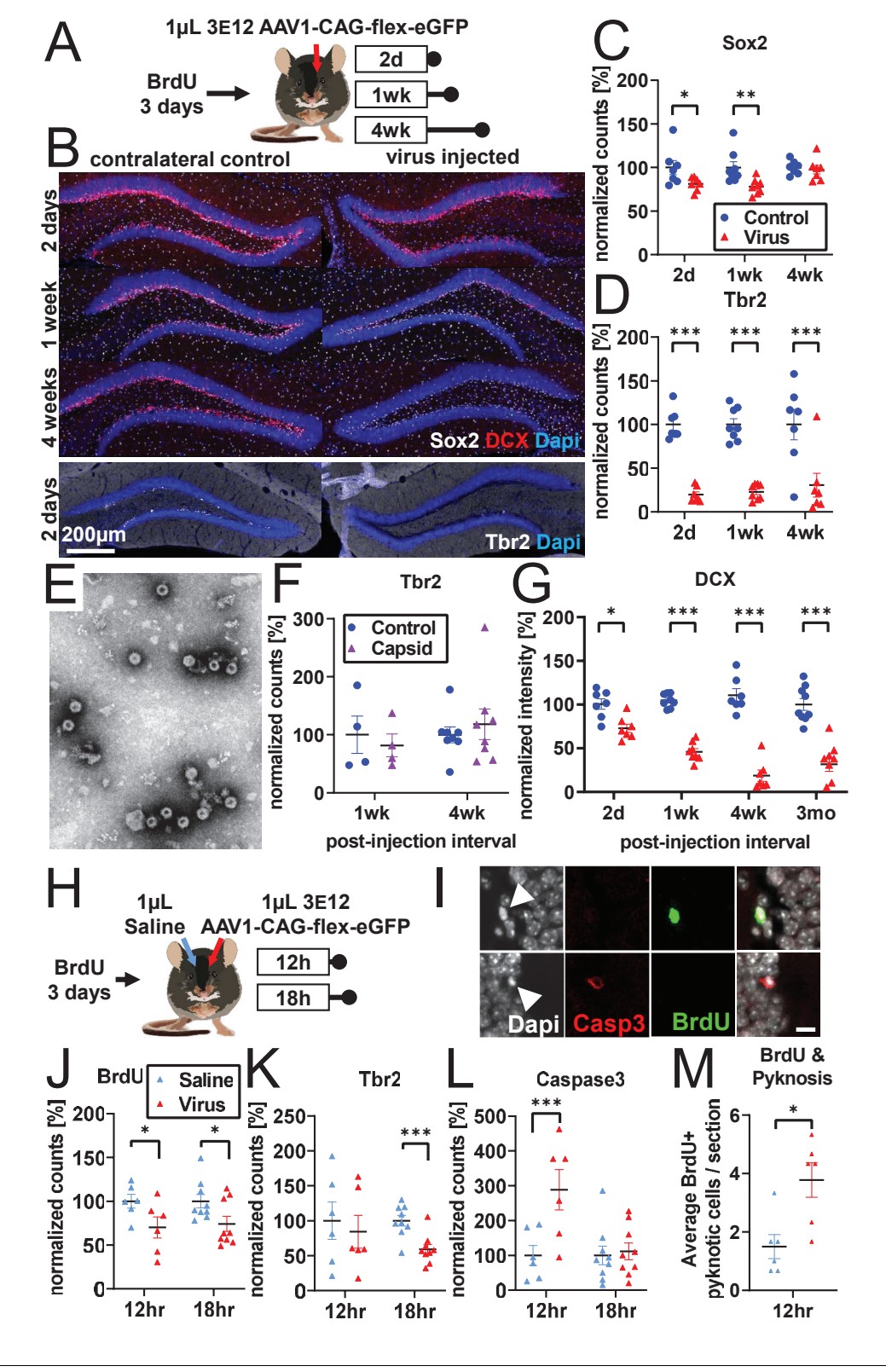

**Figure 2.** Developmental stage determines susceptibility to rAAV-induced cell loss. (A) Experimental design to assess the effect of rAAV post-injection interval on the survival of different NPC types. Following labeling with BrdU, mice are injected unilaterally with 1 μL of 3 ε12 GC/mL rAAV and sacrificed at 2 days, 1 week, and 4 weeks. (B) Representative histological staining of progenitor and immature neuronal markers Sox2 (white, upper panels), DCX (red, upper panels), and Tbr2 (white, lower panels) following rAAV injection. (C) Sox2+ neural stem cell numbers within the SGZ are reduced by ~20% 2

*Figure 2 continued on next page*

*Figure 2 continued*

days and 1 week following rAAV injection but not at 4 weeks post-injection (n = 7, 8, 7 mice per group for each time point in **C–D**). (**D**) The majority of intermediate progenitor Tbr2+ cells are lost within 2 days of rAAV injection and do not recover by 4 weeks post-injection. (**E**) Representative image of empty AAV viral particles ('empty capsid') from cryo-electron microscopy used to quantify the number of viral particles. (**F**) Tbr2+ intermediate progenitors are preserved following injection of empty viral capsid (n = 4 mice per group at 1 week, n = 8 mice at 4 weeks). (**G**) Immature neuronal marker DCX intensity shows progressive decline until complete loss at 4 weeks post-injection. DCX intensity shows no recovery at 3 months post-injection (n = 7, 8, 7, 9 mice per group for each time point). (**H**) Experimental design for acute time-line of rAAV-induced cell loss. Following labeling with BrdU, 1 μL 3 ᴇ12 GC/mL rAAV and saline control are injected into opposite sides of the DG; mice are sacrificed at 12 and 18 hr. (**I**) DNA condensation and nuclear fragmentation (pyknosis and karyorrhexis, white arrowheads) are assessed with BrdU (green), Caspase-3 activation (red), and DAPI. (**J**) BrdU+ cells show variable decline 12 hr after rAAV injection and significant decline at 18 hr relative to saline control (n = 6 mice per group at 12 hr, n = 9 mice at 18 hr for **J–M**). (**K**) Tbr2+ intermediate progenitors show significant decline by 18 hr following rAAV injection. (**L**) Caspase-3+ apoptotic cells were increased relative to saline-injected controls at 12 hr. (**M**) BrdU+ cells exhibit a significant increase in pyknosis 12 hr after rAAV injection. All data are presented as mean ± s.e.m, significance reported as: *p<0.05, **p<0.01, ***p<0.001.

The online version of this article includes the following source data and figure supplement(s) for figure 2:

**Source data 1.** Source data for *Figure 2*.
**Figure supplement 1.** Cell-type-specific cell loss.
**Figure supplement 1—source data 1.** Source data for *Figure 2—figure supplement 1*.
**Figure supplement 2.** Inflammation and cell loss.
**Figure supplement 2—source data 1.** Source data for *Figure 2—figure supplement 2*.

*1A–F*). A majority of Sox2+ cells remaining 1 week after rAAV infection are GFP positive, indicating that these cells survive despite the vast majority being infected by rAAV (*Figure 2—figure supplement 1A,F*; 75.6 ± 11.5%). However, only a minority of surviving DCX+ cells expressed GFP (*Figure 2—figure supplement 1B,F*; 35.9 ± 12.3%), and of the few remaining Tbr2 cells, few expressed GFP (*Figure 2—figure supplement 1F*; 7.1 ± 10.7%), suggesting a survivorship bias for cells not infected by rAAV when examining populations with substantial loss. Overall, we observed an increasing percentage of GFP-expressing cells with increasing survival across the three populations, suggesting that variation in viral tropism is not likely to explain the measurable differences in AAV toxicity (*Figure 2—figure supplement 1F*).

These findings suggest that, despite an initial sensitivity of some Sox2+ cells to rAAV transduction, this largely quiescent neural progenitor pool remains mostly intact. Instead, a rapid loss of proliferating Tbr2+ intermediate progenitors by 2 days drives much of the rAAV-induced toxicity, including the progressive loss of the DCX+ population that is observed as these cells differentiate into mature neurons and decline in number over time. To test the effect of increasing the size of the proliferating NPC population on rAAV-induced toxicity, we housed mice in an enriched environment with running wheels (*van Praag et al., 1999*, schematic in *Figure 2—figure supplement 1H*) Nearly all of the BrdU+-dividing cells gained from enrichment and exercise were eliminated by rAAV ($F_{housing}(1,14)=8.7$, p<0.01; $F_{treatment}(1,14)=124.0$, p<0.001; $F_{housing \times treatment}(1,14)=9.4$, p<0.01; home cage: −79.5 ± 13.9%, p<0.001; enriched environment: −139.9 ± 13.9%, p<0.001; *Figure 2—figure supplement 1I*). Tbr2+ cells were unaffected by enrichment and eliminated by rAAV regardless of housing type ($F_{housing}(1,14)=0.8$, n.s.; $F_{treatment}(1,14)=147.1$, p<0.001; $F_{housing \times treatment}(1,14)=0.5$, n.s.; *Figure 2—figure supplement 1J*). These results indicate that proliferating cells are the primary target of the virus and that environmental enrichment is insufficient to prevent cell loss.

Given the extensive and rapid loss of BrdU+ (*Figure 1B,C*) and Tbr2+ (*Figure 2B,D*) cells, we designed an acute time-course experiment to determine the mechanism of rAAV-induced cell loss (schematic *Figure 2H,I*). Following labeling with BrdU, animals were injected with 1 μL of 3ᴇ12 rAAV into unilateral dorsal DG and 1 μL saline into the contralateral DG to control for the acute effect of surgery- and injection-induced inflammation and tissue damage. rAAV-injected DGs already showed a modest decrease in BrdU+ cells at 12 and 18 hr relative to their contralateral saline-injected control ($F_{treatment}(1,13)=13.9$, p<0.01; $F_{time}(1,13)=0.04$, n.s.; $F_{treatment \times time}(1,13)=0.07$, n.s.; *Figure 2J*). The same decrease was seen in Tbr2+ cells ($F_{treatment}(1,13)=16.5$, p<0.001.; $F_{time}(1,13)=0.4$, n.s.; $F_{treatment \times time}(1,13)=3.3$, n.s.; *Figure 2K*). Cell loss was accompanied by an increased number of Caspase-3+ apoptotic cells relative to saline-injected controls at 12 hr ($F_{treatment \times time}(1,13)=21.2$, p<0.001; 12 hr treatment: +188.6 ± 29.8%, p <0.001; 18 hr treatment: 11.7 ± 24.3%, n.s.; *Figure 2L*).

Therefore, we determined that 12 hr would be a suitable time point to investigate nuclear changes in dying cells, before extensive cell loss had occurred. Condensed and fragmented chromatin (pyknosis and karyorrhexis) was identified in conjunction with BrdU and Caspase-3 (*Figure 2I*). Although the increase in total pyknotic and karyorrhexic cells in rAAV-injected DG was short of significant (p=0.058), (*Figure 2—figure supplement 1K*), a significant increase in pyknosis was seen in BrdU+ proliferating cells (2.3 ± 0.7 additional cells/section, p<0.05; *Figure 2M*). Pyknotic cells were more likely to be Caspase-3+ following rAAV injection relative to saline controls (7.7 ± 1.4 additional cells/section, p<0.01; *Figure 2—figure supplement 1L*), though BrdU+ Caspase-3+ pyknotic cells were particularly rare (n=4 of 887 cells, all in rAAV-injected DG). Taken together, these findings suggest that rAAV increases programmed cell death of dividing and recently divided cells in the DG.

Both systemic and local inflammation are known to negatively impact adult neurogenesis (*Ekdahl et al., 2003*; *Monje et al., 2003*) Therefore, we investigated whether rAAV-induced cell loss could be explained by inflammation resulting from rAAV infection. Variable inflammatory responses were observed with different viral preparations, particularly around the intermediate time point of 2 weeks (*Figure 1—figure supplement 1A*). However, rapid (<48 hr) loss of NPCs (*Figure 2*) occurred independent of expression of the microglial marker Iba1, which did not increase until 4 weeks post-injection ($F_{treatment \times time}(2,19)$=54.6, p<0.001; 2 days:18.6 ± 10.3%, n.s.; 1 week: −9.0 ± 9.7%, n.s.; 4 weeks: 132.4 ± 10.34%, p<0.001; *Figure 2—figure supplement 2A,B*). No obvious change in microglial morphology was observed at 2 days or 1 week relative to contralateral control (*Figure 2—figure supplement 2A*). At 4 weeks, microglia exhibited an amoeboid morphology, indicative of active inflammation. Similarly, expression of the astrocyte marker GFAP was unchanged in the SGZ and hilus at 2 days post-injection, slightly increased at 1 week, and greatly increased at 4 weeks ($F_{treatment \times time}(2,19)$=91.0, p<0.001; 2 days: 21.6 ± 8.7%, n.s.; 1 week: 25.2 ± 8.1%, p<0.05; 4 weeks: 165.5 ± 8.7%, p<0.001; *Figure 2—figure supplement 2C,D*). Ongoing inflammation within the neurogenic niche at the 4 week time point could exert some level of chronic toxicity on mature cells at this time point despite there being no observable loss of mature BrdU+ 8-week-old cells 1 week after viral injection (*Figure 1B*). To further rule out a bystander effect that would result from inflammation or other non-cell-intrinsic processes, we injected 30 nL of 5 E12 gc/mL AAV1-Syn-NES-jRGECO1a into the DG. Histology demonstrated incomplete loss of DCX labeling that faithfully followed the boundaries of transgene expression even at this small volume, with DCX cells within the area of viral spread lost and uninfected cells located microns away from those expressing jRGECO1a spared (*Figure 2—figure supplement 2E*). These findings suggest that AAV-induced toxicity may be cell autonomous and is unlikely to be mediated by astrocyte- or microglia-activated immune responses or by inflammatory signals and other indirect or local changes within the niche. Additionally, we found that rAAV-induced toxicity was not altered in Sting knockout mice and, therefore, likely not dependent on foreign nucleic acid detection through Sting-mediated pathways (BrdU: −90.2% ± 14.2, p<0.001; Tbr2: −88.7 ± 12.9%, p<0.001; *Figure 2—figure supplement 2F*).

## rAAV induces toxicity in NPCs in vitro

To further explore whether rAAV-mediated toxicity is cell-autonomous, we developed an in vitro assay to study rAAV-induced elimination of NPCs. Mouse NPCs were plated, administered rAAV with a multiplicity of infection (MOI) of 1 E4–1 E7 or phosphate-buffered saline (PBS) control, and chronically imaged to examine cell survival and proliferation. Dose-dependent inhibition of NPC proliferation and cell death was most profound in NPCs infected with rAAV 1 E7 MOI and moderate in NPCs infected with 1 E6 MOI. Within 24 hr of rAAV application, NPCs infected with 1 E7 MOI had ceased to proliferate, whereas application of 1 E6 MOI resulted in comparatively slower proliferation compared to PBS control. Infections with 1 E5 and 1 E4 MOI were nearly indistinguishable from PBS control ($F_{treatment}(4,665)$=1039, p<0.001; $F_{time}(18,665)$=1341, p<0.001; $F_{treatment \times time}(72,665)$=52.7, p<0.001; *Figure 3A,C*). Cell death, visualized by permeability to propidium iodide, also showed a dose-dependent increase. NPCs infected with 1 E7 MOI showed the most significant increase in cell death, whereas NPCs infected with 1 E6 MOI showed an intermediate increase. NPCs infected at 1 E5 and 1 E4 MOI were indistinguishable from PBS control ($F_{treatment}(4,665)$=9256, p<0.001; $F_{time}(18,665)$=1127, p<0.001; $F_{treatment \times time}(72,665)$=323.3, p<0.001; *Figure 3B,C*).

We then examined whether the minimum components of the AAV genome required for viral encapsulation, the 145 bp ITRs, were sufficient to induce cell death as previously reported in embryonic stem cells (*Hirsch et al., 2011*). NPCs were electroporated with 'high' (5 E6 copies/cell) and

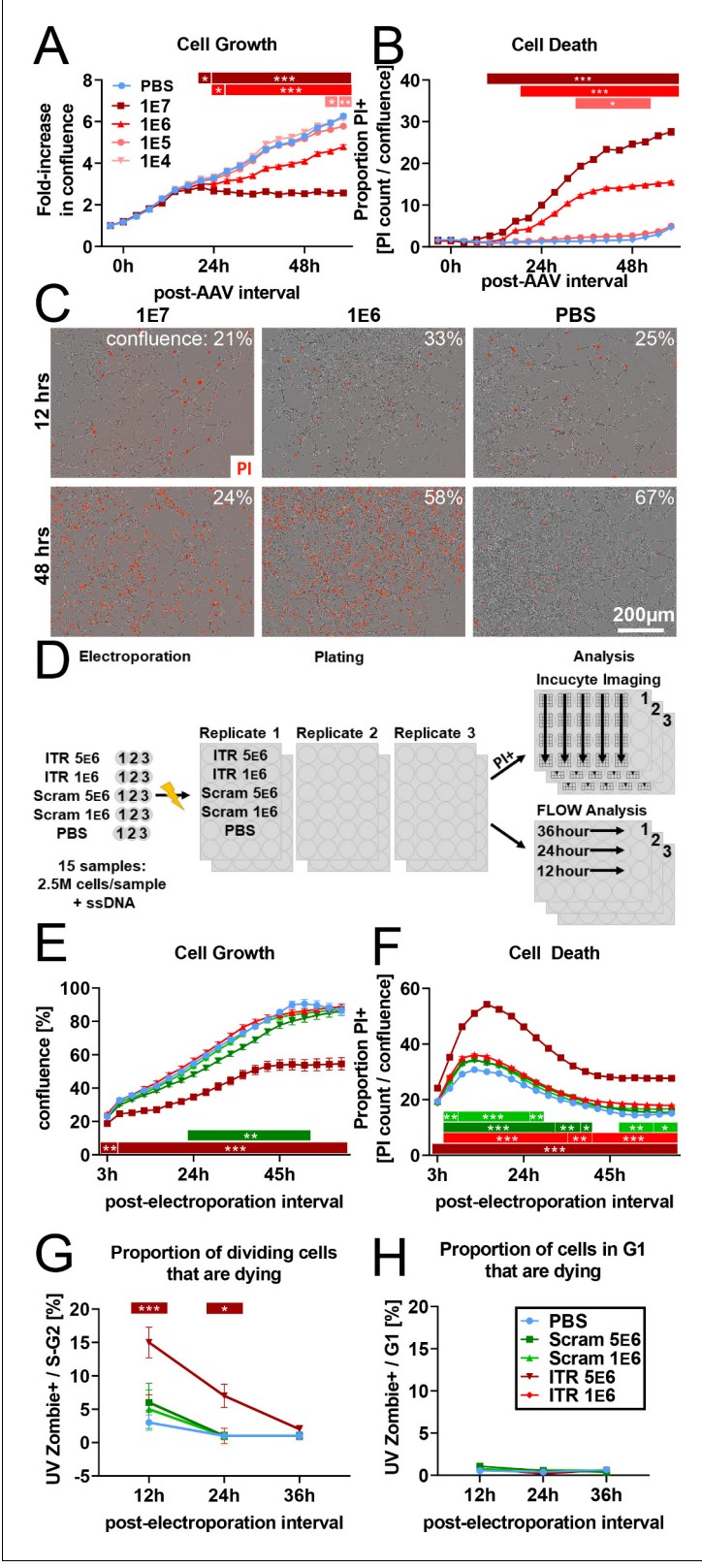

**Figure 3.** rAAV induces toxicity in NPCs in vitro. (**A**) Dose-dependent inhibition of NPC proliferation by rAAV; initial multiplicity of infectivity (MOI) of 1 ᴇ7 viral particles/cell arrests mNPC proliferation by 24 hr. MOI of 1 E6 results in slower proliferation relative to PBS control, MOI of 1 ᴇ5 and lower are indistinguishable from PBS control (n = 8 per group). (**B**) Dose-dependent rAAV-induced death; MOI of 1 ᴇ7 and 1 ᴇ6 result in increased proportion of

*Figure 3 continued on next page*

*Figure 3 continued*

propidium iodide+ NPCs (n = 8 per group). (C) Representative images showing confluence (brightfield) and propidium iodide penetration (red) into NPCs 12 and 48 hr post-viral transduction for MOI of $10^7$, $10^6$, and for PBS control. (D) Experimental design for 145 bp ssDNA AAV ITR electroporation. Mouse NPCs are electroporated with 5 ᴇ6 or 1 ᴇ6 copies of 145 bp ssDNA AAV ITR or scrambled ITR sequence control per cell and plated for FLOW analysis time course or treated with propidium iodide for imaging time course on Incucyte S3. Statistical significance shown as post hoc Tukey's multiple comparison test relative to PBS (also see *Supplementary file 1*). (E) Electroporation of 5 ᴇ6 ITR is sufficient to result in cell loss within hours of electroporation and arrest of proliferation by 40 hr. 5 ᴇ6 scrambled ITR shows slight decrease in confluence relative to 1 ᴇ6 scrambled ITR; doses of 1 ᴇ6 ITR and PBS control are indistinguishable from 1 ᴇ6 scrambled ITR (n = 3 per group). (F) Electroporation of ITRs is sufficient to induce greater levels of cell death at higher concentration of 5ᴇ6 copies/cell. (G) FLOW analysis demonstrates dose-dependent effect of rAAV ITR on replicating NPCs whereby cells electroporated with 5 ᴇ6 ITR in S- and G2- phase are dying and permeable to UVZombie at 12 hr post-electroporation. (H) NPCs in G1 represent the vast majority of cells and are not substantially dying or permeable to UVZombie, regardless of treatment. All data are presented as mean ± s.e.m, significance reported as: *p<0.05, **p<0.01, ***p<0.001. The online version of this article includes the following source data and figure supplement(s) for figure 3:

**Source data 1.** Source data for *Figure 3*.
**Figure supplement 1.** FLOW Cytometry gating strategy for analysis of AAV ITR-induced cell death during cell cycle.
**Figure supplement 1—source data 1.** Source data for *Figure 3—figure supplement 1*.

'low' (1 ᴇ6 copies/cell) doses of 145 bp rAAV2 ITR ssDNA, scrambled ITR sequence control, or PBS and plated for imaging (as above) or for FACS analysis (schematic in *Figure 3D*, gating strategy in *Figure 3—figure supplement 1*). In the high-dose ITR condition, NPCs were significantly decreased by 6 hr post-electroporation ($F_{treatment}(4,200)=630.2$, p<0.001; $F_{time}(19,200)=635.0$, p<0.001; $F_{treatment \times time}(76,200)=6.0$, p<0.001; 6 hr ITR 5 ᴇ6 vs PBS: −8.1% ± 2.5, p<0.05) and had ceased expansion by 40 hr (*Figure 3E*). Low-dose ITR and low-dose scramble groups were indistinguishable from PBS. Although a transient decrease in the high-dose scrambled condition relative to PBS was observed, this decrease was minimal compared to the effect of high-dose ITR (*Figure 3E* and *Supplementary file 1*). The number of dying propidium iodide+ cells increased in all groups in the first 24 hr following electroporation ($F_{treatment}(4,200)=3729$, p<0.001; $F_{time}(19,200)=1219$, p<0.001; $F_{treatment \times time}(76,200)=19.3$, p<0.001; *Figure 3F*), with the proportion of dying cells decreasing as confluence increased during the experiment. This proportion was substantially greater in the high ITR condition relative to PBS (*Figure 3F* and *Supplementary file 1*). Both low- and high-dose scrambled groups had a slight increase in cell death relative to PBS that was minimal compared to the effect of high-dose ITR. FACS analysis at 12, 24, and 36 hr showed the proportion of cells in S/G2 phase that were dying (UVZombie+) was greatly increased at 12 and 24 hr in the high ITR condition, but not in the other experimental groups relative to PBS control ($F_{treatment}(4,30)=9.5$, p<0.001; $F_{time}(2,30)=23.3$, p<0.001; $F_{treatment \times time}(8,30)=2.2$, p<0.05; 12 hr ITR 5ᴇ6 vs PBS +12.0 ± 2.0%, p<0.001, *Figure 3G*). The proportion of non-replicating cells that were dying was <1% in all groups (*Figure 3H*).

## AAV retro serotype permits studies of DGC activity in vivo without ablating adult neurogenesis

To determine the functional consequence of AAV-induced ablation of neurogenesis on DG activity, animals were injected with 1 μL of 3 ᴇ12 gc/mL, 1 ᴇ12 gc/mL, or 3 ᴇ11 gc/mL rAAV and exposed 4 weeks later to a novel environment (NE) prior to sacrifice. DGC expression of the immediate early gene cFOS was used to quantify the effect of AAV on DG activity. Consistent with previous studies examining the effects of manipulating adult neurogenesis on DG activity (*Ikrar et al., 2013*), mature DGC cFOS activation showed an inverse relationship with the level of adult neurogenesis ($F_{treatment \times time}(2,20) = 11.4$, p<0.001; *Figure 4A,B*). Injection with 1 μL 3 ᴇ12 gc/mL rAAV, which ablates over 80% of BrdU+ cells (*Figure 1D*), resulted in the largest increase in mature DGC cFOS activation (81.6±13.6 additional cells per section, p<0.001); 1 μL 1 ᴇ12 gc/mL rAAV injection resulted in a moderate and more variable increase (40.0 ± 13.6 additional cells per section, p<0.05), and 1 μL 3 ᴇ11 gc/mL rAAV injection demonstrated no significant change on average (-13.3 ± 14.5 additional cells

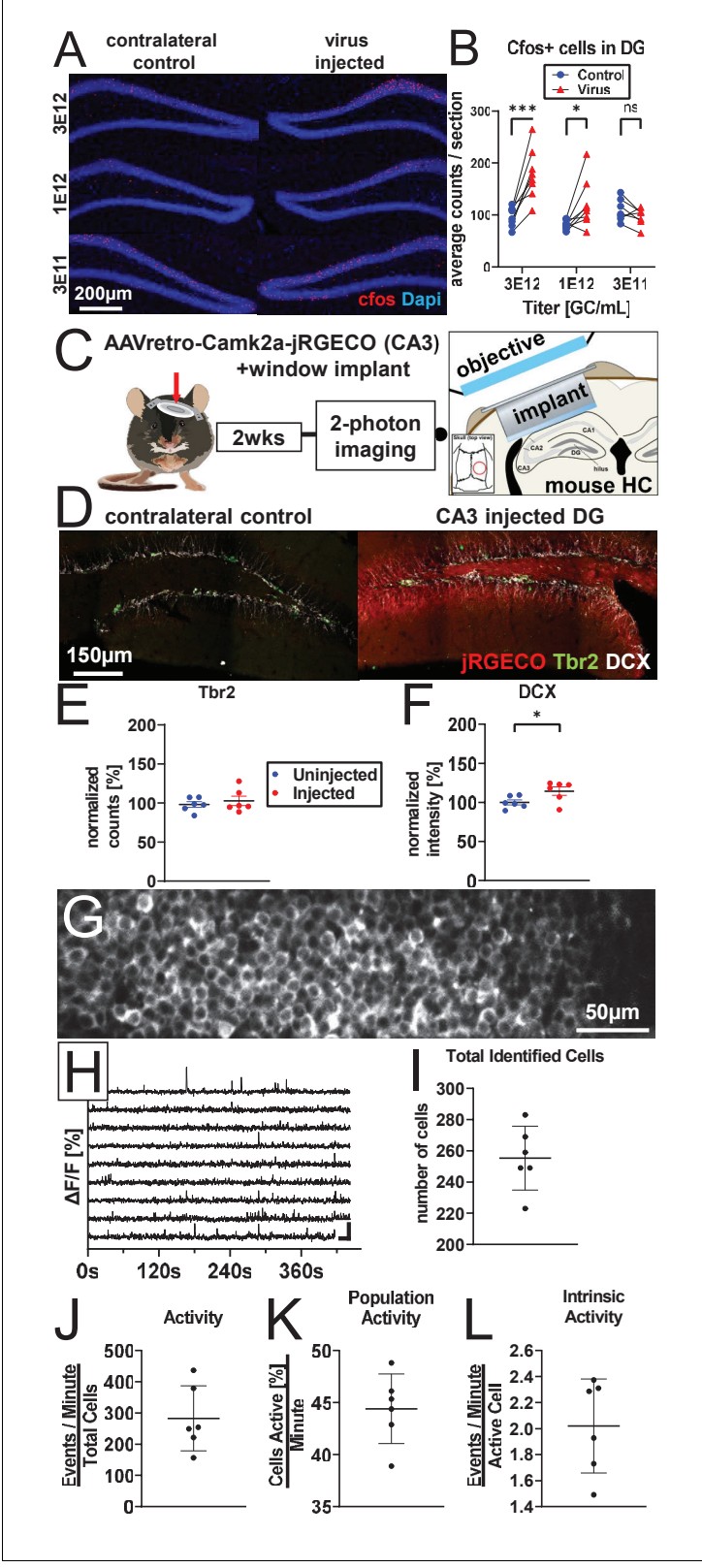

**Figure 4.** AAV retro serotype permits studies of DGC activity in vivo without ablating adult neurogenesis. (**A**) Representative images showing cFos and Dapi used for quantification. (**B**) Mature DGCs are hyperactive following rAAV-induced cell loss in a dose-dependent manner for injected titers between 3E12 and 3E11. abDGC knockdown efficiency is significantly correlated with cFOS activation in mature DGCs (*Figure 4—figure supplment 1A*). (**C**)
*Figure 4 continued on next page*

*Figure 4 continued*

Experimental design for two-photon imaging of DG utilizing AAV retro. Eight hundred nanoliters of 3 ᴇ12 GC/mL AAVretro-CaMKIIa-jRGECO1a is injected into CA3, a cranial window is implanted, and mice undergo two-photon calcium imaging 2 weeks later (adapted from *Gonçalves et al., 2016*). (D) Representative images show Tbr2+ and DCX+ cells are intact in animals injected with AAV retro in CA3. (E) Quantification of Tbr2+ intermediate progenitors demonstrating adult neurogenesis is intact in AAV retro-injected animals. (F) Quantification of DCX staining demonstrates adult neurogenesis is intact in AAV retro-injected animals. (G) Representative field of view maximum projection for 2-photon calcium imaging showing cytoplasmic expression of jRGECO1a in >200 DGCs within a field of view. (H) Representative calcium traces of 10 randomly selected neurons from the same animal shown above. (I) Total number of identified DGCs in each DG is similar across animals. (J–L) Total activity of these cells was demonstrated to be sparse, with approximately one quarter of all cells active in any given minute while imaging, and only a few calcium transients per active cell a minute. Data are presented as mean ± s.e.m when comparing between groups in C–D and mean ± s.d. when describing variability within groups in I–L, significance reported as: *p<0.05, **p<0.01, ***p<0.001.

Gonçalves et al.

. Graphic in Panel C adapted from Figure 1A in *Gonçalves et al., 2016*. This image is not covered by the CC-BY 4.0 licence and further reproduction of this panel would require permission from the copyright holder.

The online version of this article includes the following source data and figure supplement(s) for figure 4:

**Source data 1.** Source data for *Figure 4*.

**Figure supplement 1.** Calcium imaging following 2 week knockdown of adult neurogenesis.

**Figure supplement 1—source data 1.** Source data for *Figure 4—figure supplement 1*.

per section, n.s.). cFOS activation and loss of BrdU+ cells were significantly correlated (slope = 0.26, $R^2$=0.59, p<0.001; *Figure 4—figure supplement 1A*). These results demonstrate that, while viral titers producing severe ablation of neurogenesis have the most severe effect on DG activity, titers that do not completely eliminate neurogenesis still have an impact on the network.

Given the importance of adult neurogenesis in regulating population activity in the DG, we sought a method that would permit two-photon calcium imaging of in vivo network activity within the DG without ablating neurogenesis. AAV retro is a designer AAV variant capsid optimized to be taken up by axonal projections (*Tervo et al., 2016*). We used AAV retro (AAVretro-CaMKIIa::NES-jRGECO1a) to deliver jRGECO1a to DGCs in a retrograde fashion by injecting virus into the dorsal CA3, where their axons ('mossy fibers') terminate. This delivery method allowed us to avoid infecting immature DGCs whose mossy fiber projections do not reach CA3 until after 2 weeks of age, a time point when these neurons also demonstrate decreased susceptibility to rAAV-induced toxicity (*Figure 1B*). Following viral injection, a cranial window was implanted to permit visualization of DGC activity via two-photon calcium imaging (schematic in *Figure 4C*).

Histology performed following two-photon calcium imaging confirmed that adult neurogenesis was intact 2 weeks after injection relative to the uninjected contralateral hippocampus (Tbr2: +4.7 ± 7.0%, t(5)=0.7, n.s. *Figure 4D,E*; DCX +14.3 ± 6.3%, t(5)=4.3, p<0.01, *Figure 4D,F*). Spontaneous calcium activity was recorded in DGCs and quantified in these mice under head fixation on a cylindrical treadmill in the dark (*Figure 4G,H*). A similar number of DGCs was identified across animals (255.3 ± 20.5 cells/DG; *Figure 4I*). Total activity of these cells was demonstrated to be sparse (282.9 ±104.5 events/minute; *Figure 4J*). Approximately one quarter of the cells were active in any given minute during imaging (24.3 ± 7.6% cells active/minute; *Figure 4K*), with a small number of calcium transients per active cell observed per minute (2.0 ± 0.4 events/active cell/minute; *Figure 4L*). The limited activity of DGCs measured in our study with AAV retro is consistent with previous imaging studies in the DG (*Danielson et al., 2016*; *Pilz et al., 2016*). Due to the extensive inflammation that arises 4 weeks after viral injection (*Figure 1—figure supplement 1A*, *Figure 2—figure supplement 2A–D*), we were unable to image the DG at 4 weeks post-viral injection, when abDGCs are highly active and known to contribute to DG activity (*Figure 4A*). Despite this limitation, a small suppressive effect of AAV1 injection on DG activity was already evident by 2-photon imaging 2 weeks post injection (*Figure 4—figure supplement 1*), when DG imaging is often initiated (*Danielson et al., 2016*; *Hainmueller and Bartos, 2018*; *Pilz et al., 2016*). No change in cFOS expression due to AAV1 was observed at this earlier time point (*Figure 4—figure supplement 1K*). Collectively, these findings demonstrate that (1) delivery of AAV directly into the DG ablates neurogenesis and

influences activity in this network, particularly at time points when abDGCs make a strong contribution to DG function, and (2) AAV retro enables delivery of transgenes to the DG, permitting studies of DGC function in vivo while leaving adult neurogenesis intact.

## Discussion

### A developmental window for sensitivity to rAAV-induced toxicity

We demonstrate that adult murine NPCs and immature neurons up to approximately 1 week of age are eliminated by rAAV in a dose-dependent fashion (*Figure 1*). The doses demonstrated to ablate neurogenesis are within or below the range of experimentally relevant titers commonly injected into the mouse DG, 1.5 E12 to 3.6 E13 gc/mL (*Anacker et al., 2018*; *Castle et al., 2018*; *Danielson et al., 2016*; *Danielson et al., 2017*; *Gong and Zhou, 2018*; *Hashimotodani et al., 2017*; *Hayashi et al., 2017*; *Kaspar et al., 2002*; *Kirschen et al., 2017*; *Liu et al., 2012*; *McAvoy et al., 2016*; *Ni et al., 2019*; *Pilz et al., 2016*; *Ramirez et al., 2015*; *Raza et al., 2017*; *Redondo et al., 2014*; *Senzai and Buzsáki, 2017*; *Swiech et al., 2015*; *Zetsche et al., 2017*). This rAAV-induced cell death is rapid and persistent; BrdU-labeled cells and Tbr2+ intermediate progenitors begin to die within 12–18 hr post-injection and are eliminated by 48 hr (*Figure 2*). Under physiological conditions, the Tbr2+ population replenishes DCX+ progenitors and immature neurons, which can retain expression of the DCX protein in the mouse for more than 3 weeks post-mitosis (*Kempermann et al., 2003*). Upon administration of rAAV, many of the DCX+ neurons are postmitotic and initially spared but are not replenished by the ablated Tbr2+ population, explaining the delayed and progressive loss of the DCX+ pool in response to rAAV infection. The immature neuron population showed no evidence of recovery when assessed several months post-injection (*Figure 2G*). Interestingly, the total number of Sox2+ cells, composed of Type 1 and 2a NPCs, was relatively unaffected by rAAV, consistent with this population being largely quiescent (*Figure 2B,C*). The fact that Sox2+ cells were mostly preserved in our experiments might explain why Type I cells are visualized in studies utilizing a large range of AAV titers, including titers greater than those reported here (*Crowther et al., 2018*; *Kotterman et al., 2015*; *Ojala et al., 2018*; *Pulicherla et al., 2011*; *Song et al., 2012*).

In vitro application of rAAV or electroporation of AAV2 ITRs is sufficient to induce arrest of proliferation and cell death, pointing toward a cell autonomous process (*Figure 3*). This is consistent with our findings that the timing of rAAV-induced apoptosis and cell loss (hours to days) is not commensurate with the time course or spatial extent of inflammation (*Figure 2—figure supplement 2A–D*) and is independent of exposure to empty capsids, which lack ITRs (*Figures 1B* and *2F*, *Figure 1—figure supplement 1C*, *Figure 2—figure supplement 1G*). Analysis with FACS demonstrated that administration of high-dose ITR oligonucleotides resulted in a disproportionate loss of dividing cells (*Figure 3G–H*). To a large extent, this finding seems to recapitulate in vivo experiments, which demonstrate that a developmental window exists where actively dividing Type 2 NPCs and recently postmitotic immature neurons are sensitive to rAAV-induced cell death, whereas rarely dividing neural stem cells and mature abDGCs flanking this window are significantly less affected. However, future in vivo studies using techniques that permit viral free delivery of large oligomers, approaching the size of ITRs, are required to establish sufficiency of ITR toxicity in vivo.

Note that the persistence of Sox2 expression in dividing Type 2a progenitors may account for the initial modest decrease seen in the number of Sox2+ cells (*Figure 2B,C*, *Gonçalves et al., 2016*; *Kempermann et al., 2015*). Why these cells do not replenish the Tbr2+ pool following rAAV infection remains unknown. One possibility is that Type I cells infected with rAAV undergo delayed apoptosis upon entering the cell cycle, precluding the recovery of neurogenesis after infection. However, few Sox2+ cells were lost over the course of 4 weeks (*Figure 2C*, *Figure 2—figure supplement 1C*) and triple labeled Sox2+GFP+caspase3+ cells 1 week after viral injection were exceedingly rare. Alternatively, delayed inflammation observed in our experiments 1 month after rAAV injection could create a non-permissive environment for neurogenesis and could explain the inability of Sox2 cells to replenish the Tbr2 and DCX cell populations. However, both in vivo and in vitro experiments indicate that inflammation does not account for the initial elimination of dividing NPCs by AAV.

## rAAV as a model system for viral toxicity in the developing CNS

Infections involving a number of viruses, including cytomegalovirus (CMV), rubella, varicella-zoster, herpes simplex, human immunodeficiency virus (HIV), and Zika, have been implicated in the pathogenesis of microcephaly, the abnormal development of the cerebral cortex resulting in small head size. There is evidence that these viruses cause microcephaly through the elimination of NPC populations (*Devakumar et al., 2018*) in HIV (*Balinang et al., 2017*; *Schwartz et al., 2007*), CMV (*Luo et al., 2010*; *Teissier et al., 2014*), and Zika virus (*Garcez et al., 2016*; *Qian et al., 2016*) among others. However, the complex biology of these viruses and their neural progenitor targets has precluded elucidation of a precise mechanism, despite the association between these viruses and microcephaly being known for over 75 years (*Swan et al., 1946*). Perhaps the best studied among these is Zika virus, which, similar to rAAV, attenuates proliferation and neurogenesis in the adult mouse DG, demonstrating a marked loss of EdU+ cells following infection (*Li et al., 2016*). These findings resemble the loss of proliferating cells in brain organoids and other models of the developing nervous system in response to Zika infection (*Cugola et al., 2016*; *Garcez et al., 2016*; *Qian et al., 2016*). Collectively, these studies have focused on Zika virus' selective tropism for Sox2+ and Nestin+ NPCs over Tbr2$^+$ and other immature cell types in the brain. However, these studies typically measure the fraction of Zika-infected cells that express Sox2, Tbr2, and other immature markers, but do not track changes in the total size of these populations resulting from Zika infection. These measurements could have missed a rapid elimination of Tbr2 cells, which in our measurements was apparent within 48 hr of rAAV infection (*Figure 2D,K*). Alternatively, the proliferative capacity and thus the susceptibility of the Sox2+ population to viral infection could differ between developmental models and the adult DG, where in adult neurogenesis Sox2+ cells are largely quiescent and perhaps less sensitive to virus-induced cell death (*Wu et al., 2018*). Further studies are needed to discern the downstream events that lead to viral toxicity and whether this heterogeneous collection of viruses kills dividing NPCs through a common pathway. While a detailed mechanism for virus-induced toxicity in NPCs remains elusive, rAAV, with its exceedingly simple genome, broad tropism, and inability to replicate, offers a tractable model system to dissect the molecular events underlying this important phenomenon.

## Implications for DG and hippocampal function

Both theoretical and experimental studies indicate that the DG is involved in hippocampus-dependent behavioral pattern separation and pattern completion (*Chawla et al., 2005*; *Deng et al., 2013*; *Lacy et al., 2011*; *Leutgeb et al., 2007*; *McClelland et al., 1995*; *Treves and Rolls, 1994*). More recent studies implementing genetically encoded tools, often delivered via rAAV, provide striking evidence for the role of memory engram representations in behavioral pattern separation and completion in the DG (*Bernier et al., 2017*; *Danielson et al., 2016*; *Liu et al., 2012*; *Ramirez et al., 2015*; *Redondo et al., 2014*) and highlight the role of this circuit in affective disorders and stress responses (*Anacker et al., 2018*; *Ni et al., 2019*; *Ramirez et al., 2015*). Moreover, DG activity and computations appear to depend on the addition of abDGCs (*Clelland et al., 2009*; *Ikrar et al., 2013*; *Sahay et al., 2011*), whose net effect is to quiet activity in mature DGCs and the rest of the hippocampus (*Berdugo-Vega et al., 2020*). This inhibition on mature DGCs through either monosynaptic (*Luna et al., 2019*) or polysynaptic inhibition (*Jinde et al., 2013*; *Toni et al., 2008*) is thought to enhance pattern separation by selectively suppressing competing engrams (*Espinoza et al., 2018*; *Johnston et al., 2016*; *McAvoy et al., 2016*; *Sahay et al., 2011*). Consistent with this idea, rAAV-induced ablation of neurogenesis results in a dose-dependent increase in mature DGC activity 4 weeks after infection, as measured by immediate early gene expression (*Figure 4A,B*, *Figure 4—figure supplement 1A*).

In the majority of studies in which rAAV was injected into the DG, neurogenesis was not assessed following viral transduction (see *Danielson et al., 2016*; *Sparks et al., 2020*; *Song et al., 2012* for instances where neurogenesis is assessed). However, *Song et al., 2012* were able to observe and quantify Type I Nestin+ radial glia-like cells in the DG 4 weeks after AAV injection (other NPC or immature neuronal markers were not measured). Only ~2–6% of radial glia-like cells were dividing in these experiments, consistent with our measurements showing that Sox2+ cells at this time-point are resistant to AAV toxicity (*Figure 2*). Also, *Danielson et al., 2016* and *Sparks et al., 2020* reported in vivo calcium imaging of adult-born and mature DGCs using rAAV for delivery of

GCaMP6. In this paradigm, abDGCs were labeled via Tamoxifen administration in a Nestin-CreER x tdTomato reporter mouse 3 weeks before rAAV injection into the DG at titers of ~1 ᴇ13 gc/mL. Imaging took place 3 weeks later, when tdTomato-labeled cells were ~6 weeks of age. This paradigm would permit tdTomato-labeled abDGCs that were 3 weeks of age at the time of rAAV injection and 6 weeks old at imaging to largely escape rAAV-induced toxicity. However, the loss of abDGCs ~4 weeks old and younger and their contribution to activity in the DG might have been missed.

The retrograde labeling of DGCs by injecting AAV retro into CA3 provides an important advance for future studies of the DG, as DGCs can now be imaged and manipulated using genetic tools while leaving adult neurogenesis intact (*Figure 4*). Relatively few studies have performed calcium imaging of the DG in vivo (*Danielson et al., 2016*; *Danielson et al., 2017*; *Hainmueller and Bartos, 2018*; *Pilz et al., 2016*). Variability in experimental approaches used to deliver genetically encoded calcium indicators to DGCs and inevitable differences in segmentation routines and analysis of calcium traces make direct comparison with these published results difficult. The results described above do not differ substantially with previous findings of ~40–50% of cells being active during recording (*Danielson et al., 2016*; *Pilz et al., 2016*) and producing only a few calcium transients per minute (*Danielson et al., 2016*). However, given the ability of abDGCs to modulate activity within the DG (*Ikrar et al., 2013*; *Lacefield et al., 2012*; *Luna et al., 2019*), viral methods used to deliver calcium indicators and other genetic tools into the DG should be carefully evaluated in future studies.

## Caveats for gene therapy

Based on its stable transgene expression, low risk of insertional mutagenesis, and diminished immunogenicity, rAAV has become the most widely used viral vector for human gene therapy. More than 100 clinical trials using AAV vectors have claimed vector safety (*Choudhury et al., 2017*; *Hocquemiller et al., 2016*; *Hudry and Vandenberghe, 2019*), resulting in two FDA-approved therapies for treating genetic diseases of the CNS (*Hoy, 2019*; *Smalley, 2017*; *Mendell et al., 2017*). Despite high rates of infection among humans (*Thwaite et al., 2015*), AAV infection has not been associated with illness or pathology (*Büning and Schmidt, 2015*). However, recent reports have suggested rAAV may exhibit intrinsic toxicity in multiple tissues (*Bockstael et al., 2012*; *Flotte and Büning, 2018*; *Hinderer et al., 2018*; *Hirsch et al., 2011*; *Hordeaux et al., 2018*). Given the steep dose response of rAAV-induced cell death measured in our study, intravenous administration of rAAVs at clinically relevant titers is less likely to cross the blood-brain barrier (BBB) and reach the SGZ of the DG with sufficient MOI to ablate neurogenesis. However, the protective capacity of the BBB does not preclude AAV-induced toxicity in other progenitor and dividing cells throughout the body. Further studies are needed to characterize rAAV-induced toxicity to stem cells and progenitor cells in other tissues. Also, high MOI may reach the SGZ in clinical trials where rAAV is injected intrathecally (*U.S. National Library of Medicine, 2017*) or directly into brain tissue (*Castle et al., 2018*; *Tuszynski et al., 2015*). While attenuation of neurogenesis may occur in patients undergoing other treatments such as chemotherapy (*Hodge et al., 2008*) or radiation treatment (*Saxe et al., 2007*), the extent of ablation induced by rAAV is striking and shows no signs of recovery throughout the duration of our experiments. This study serves as an additional reminder that rAAV and other viral gene therapies may be associated with significant side effects, particularly during development. Careful consideration of viral titer, delivery method, and viral engineering should be exercised to mitigate side effects where viral therapy may substantially alleviate morbidity or extend life.

## Materials and methods

### Animal use

All animal procedures were approved by the Institutional Animal Care and Use Committees of the Salk Institute and the University of California San Diego, and all experiments were conducted according to the US Public Health Service guidelines for animal research. Wild-type male C57BL/6J mice (Jackson Laboratories stock #000664) or Sting-KO mice (Jackson Laboratories #025805, *Jin et al., 2011*), 6–7 weeks of age at the time of surgery, were used in this study. Unless otherwise noted, mice were group housed with up to five mice per cage in regular cages (14.7' L × 9.2' W ×

5.5' H, InnoVive, San Diego, CA) under standard conditions, on a 12 hr light–dark cycle, with ad libitum access to food and water. BrdU (Sigma) was administered i.p. at 50 mg/kg/day for 3 days.

## Viral injection

Mice were anesthetized with isoflurane (2% via a nose cone, vol/vol), administered with dexamethasone (2.5 mg/kg, i.p.) to decrease inflammation, and placed in a stereotaxic frame. A single injection of 1 µL of virus solution diluted in sterile saline or saline control, unless otherwise specified, was delivered to the dorsal hippocampus through stereotaxic surgery using a microinjector (Nanoject III, Drummond Science). Specifically, the difference between bregma and lambda in anteroposterior coordinates was determined. From bregma, DG injection coordinates were calculated as indicated in *Table 1*. AAV spread extended to transduce approximately ½ of the dorsal-ventral extent of the DG (*Figure 1—figure supplement 1E*).

In any surgical manipulation of the DG, including our studies, abDGCs are exposed to a variety of experimental manipulations that might affect adult neurogenesis, including anesthesia, nonsteroidal anti-inflammatory drugs, and corticosteroids (*Cameron and Gould, 1994*; *Erasso et al., 2013*; *Kim et al., 2020*; *Lehmann et al., 2013*; *McGuiness et al., 2017*; *Monje et al., 2003*; *Saaltink and Vreugdenhil, 2014*; *Schoenfeld and Gould, 2013*; *Stratmann et al., 2009*; *Stratmann et al., 2010*) utilized for animal comfort and humane experimentation. The effects of these pharmacological agents are largely accounted for by performing intra-subject comparisons, where the uninjected or saline-injected contralateral hippocampus is also exposed to these agents. Previous work indicates that these compounds have no significant effect on the development of dendritic arbors in abDGCs, which are sensitive to experience (*Gonçalves et al., 2016*).

CA3 injection coordinates were calculated as follows: anteroposterior (A/P) −1.8 mm, lateral (M/L) −1.8 mm, ventral (V/L; from dura) −1.6 mm and −2.0 mm, with 400 nL injected at each depth. Following completion of the surgery, carprofen (5 mg/kg, i.p.) and Buprenorphine SR LAB (1.0 mg/kg, s.c.) were administered for inflammation and analgesic relief. Mice were allowed to recover and then returned to their cages. The following viral vectors (i.e. plasmid, production core) were used: AAV1-CAG-GFP (Addgene 37825; Addgene), AAV1-CAG::flex-eGFP-WPRE-bGH (Addgene 51502, U Penn and Addgene), AAVretro-CaMKIIa::NES-jRGECO1a-WPRE-SV40 (Gage, Salk), AAV8-CaMKIIa::NES-jRGECO1a-WPRE-SV40 (Gage, Salk), AAV1-Syn::NES-jRGECO1a-WPRE-SV40 (Addgene 100854, U Penn), AAV8-CaMKIIa::mCherry-WPRE-bGH Addgene 114469, Salk, AAV8 capsid (University of North Carolina Viral Vector Core pXR8, Salk, see Appendix 1: Key Resources Table).

## Enriched/novel environments

Mice assigned to enriched environments (EE) were housed in regular caging and then moved to an EE cage, whereas matched home cage (HC) controls remained in regular caging. The EE cage (36' L × 36' W × 12' H) contained a feeder, two to three water dispensers, a large and a small running wheel and multiple plastic tubes and domes, and paper huts, with a 12 hr light–dark cycle. Objects in the EE cage were kept constant throughout the experiment; placement of the objects was altered only to the extent that the mice moved them within the cages. Mice were kept in EE or HC for 13

**Table 1.** DG Injection coordinates.

Injection coordinates as measured from bregma adjusted for measured distance between lambda and bregma (Λ-B): anterior–posterior (A/P), medial–lateral (M/L); and dorsoventral depth from dura (D/V).

| Λ-B [mm] | A/P [mm] | M/L [mm] | D/V [mm] |
|---|---|---|---|
| 3.0 | −1.5 | ±1.5 | −1.8 |
| 3.2 | −1.6 | ±1.55 | −1.8 |
| 3.4 | −1.7 | ±1.6 | −1.9 |
| 3.6 | −1.8 | ±1.65 | −1.9 |
| 3.8 | −1.9 | ±1.7 | −1.95 |
| 4.0 | −2.0 | ±1.75 | −2.0 |

days and injected with BrdU on the final 3 days. On the final day of BrdU, mice were also unilaterally injected with 1 µL 3 E12 gc/mL AAV1-CAG-flexGFP into the DG. Following surgery, animals were returned to EE or HC and sacrificed 2 days post-injection. Mice that received novel environment exposure for cFOS activation experiments remained in HC until the time of exposure and were then transferred to EE cages (as described above) for 15 min. Animals were sacrificed and brain tissue collected 1 hr after exposure.

## Cranial window placement

For two-photon calcium imaging experiments, ~1 hr after receiving viral injections as described above, a ~3 mm diameter craniotomy was performed, centered around the DG viral injection site. The underlying dura mater was removed and the cortex and corpus callosum were aspirated with a blunt tip needle attached to a vacuum line. Sterile saline was used to irrigate the lesion and keep it free of blood throughout the surgery. A custom 3 mm diameter, 1.4 mm deep titanium window implant with a 3 mm glass coverslip (Warner Instruments) bottom was placed on the intact alveus of the hippocampus. The implant was held in place with UV-cured dental adhesive (Kerr Dental, Optibond All-In-One) and dental cement (Lang Dental, Ortho-Jet). A small custom titanium head bar was attached to the skull to secure the animal . Following completion of the surgery, carprofen (5 mg/kg, i.p.) and Buprenorphine SR LAB (1.0 mg/kg, s.c.) were administered (as previously mentioned, animals received 1 dose of each at the end of the final surgery) for inflammation and analgesic relief. Mice were allowed to recover and then returned to their cages. We have previously found that surgical implant and imaging procedures do not affect adult neurogenesis (see *Gonçalves et al., 2016*, Fig. S2.8 and S9).

## Two-photon calcium imaging of DG activity

Mice were acclimated to head fixation beginning 1 week after surgery. At time of imaging, each mouse was secured to a goniometer-mounted head-fixation apparatus and a custom-built laser alignment tool was used to level the plane of the cranial window coverslip perpendicular to the imaging path of the microscope objective. Imaging of dorsal DG was performed with a two-photon laser scanning microscope (MOM, Sutter Instruments) using a 1070 nm femtosecond-pulsed laser (Fidelity 2, Coherent) and a 16× water immersion objective (0.8 NA, Nikon). Images were acquired using the ScanImage software implemented in MATLAB (MathWorks). Imaging sessions were performed intermittently from 10 to 18 dpi to determine optimal viral expression and imaging window. Analyzed activity videos were acquired at ~14 dpi in successive 5 min intervals (512 × 128 pixels; ~3.91 Hz).

## Analysis and quantification of calcium activity

Custom software was written in Matlab (available at https://github.com/shtrahmanlab/CaImagingDataElife2021.git copy archived at swh:1:rev:4909eb98ba002b29525f8dfdd9699012e6880d76 *Johnston et al., 2021*) to extract neuronal activity from two-photon calcium imaging videos. Calcium traces were extracted by first performing image stabilization for each video using a rigid alignment, maximizing the correlation coefficient between each frame of the movie with an average reference frame constructed from 20 to 30 frames acquired when the mouse was not running. Alignment was further improved using a line-by-line alignment. Portions of the movie with excessive movement artifact, defined by cross-correlation coefficients below a defined threshold, were discarded. Mice were not trained or rewarded for running on the treadmill and thus were stationary during the majority of the presented calcium imaging data. Automated cell segmentation was achieved by scanning a ring shape of variable thickness and size across a motion corrected reference image. When the cross-correlation metric exceeded a user-adjustable threshold, a circular shaped ROI was generated and the signal extracted. User input was then taken for each video to remove a small number of false positives and labeled cells that evaded automatic classification. Once each cell was labeled and the intensities were recorded, the baseline fluorescence (F) was fit to an exponential curve to eliminate photo-bleaching effects. The change in fluorescence ($\Delta F$) over the baseline fluorescence (F) was then calculated to yield %$\Delta F$/F. Spiking-related calcium events for each cell were defined as fluorescence transients whose amplitude exceeded seven standard deviations of the negative fluctuations of the %$\Delta F$/F trace. Active cells were defined as having at least one spike during a single movie.

## Tissue collection

Mice were deeply anesthetized with ketamine and xylazine (130 mg/kg, 15 mg/kg; i.p.) and perfused transcardially with 0.9% PBS followed by 4% paraformaldehyde (PFA) in 0.1 M phosphate buffer (pH 7.4). Brains were dissected and post-fixed in 4% PFA overnight and then equilibrated in 30% sucrose solution.

## Immunohistochemistry

Fixed brains were frozen and sectioned coronally on a sliding microtome at 40 µm thickness, spanning the anterior–posterior extent of the hippocampus, and then stored at −20°C until staining. Brain sections were blocked with 0.25% Triton X-100 in TBS with 3% horse serum and incubated with primary antibody in blocking buffer for 3 nights at 4°C. Sections were washed and incubated with fluorophore-conjugated secondary antibodies for 2 hr at RT. DAPI was applied in TBS wash for 15 min at RT. Sections were washed and mounted with PVA-Dabco or Immu-Mount mounting media. For BrdU staining, brain sections were washed 3× in TBS for 5 min, incubated in 2N HCL in a 37°C water bath for 30 min, rinsed with 0.1M Borate buffer for 10 min at RT, washed 6× in TBS for 5 min, and then the above staining procedure was followed.

Primary antibodies used were rat αBrdU (OBT0030, Accurate; NB500-169, Novus; AB6326, Abcam), rabbit αcleaved-CASPASE3 (9661, Cell Signaling), goat αCFOS (sc-52-G, Santa Cruz), rabbit αCFOS (226003, Synaptic Systems), goat αDCX (sc-8066, Santa Cruz), guinea pig αDCX(AB2253, Millipore), chicken αGFAP (AB5541, Millipore), chicken αGFP (GFP-1020, Aves Labs), rabbit αPROX1 (ab101851, Abcam), rabbit αSOX2 (2748, Cell Signaling), rabbit αTBR2 (ab183991, Abcam), and rat αSOX2 (14981182, Invitrogen). Secondary antibodies used were donkey αchicken-AlexaFlour647 (703-605-155), donkey αchicken-AlexaFluor488 (703-545-155), donkey αrat-AlexaFluor647 (712-605-153), donkey αrabbit-Cy5 (711-175-152), donkey αrabbit-Cy3 (711-165-152), donkey αrabbit-AlexaFluor488 (711-545-152) donkey αguinea pig-AlexaFluor488 (706-545-148), donkey αguinea pig-Cy3 (706-165-148), donkey αguinea pig- AlexaFlour647 (706-605-148), donkey αgoat – AlexaFlour647 (705-175-147), donkey αgoat – Cy3 (705-165-147), and donkey αgoat AlexaFlour488 (705-545-147) – (Jackson Immuno Research Laboratories).

## Histology acquisition and analysis

Images for analysis of neurogenesis and inflammation markers were acquired using a Zeiss laser scanning confocal microscope (LSM 710, LSM 780, or Airyscan 880) using a 20× objective or an Olympus VS-120 virtual slide scanning microscope using a 10× objective. For confocal images, Z-stacks were obtained through the entirety of the DGC layer, tiles were stitched using Zen software (Zeiss), and images were maximum projected for quantification. Slide scanner images were obtained from a single plane. For markers quantified by cell counts (BrdU, TBR2, SOX2, CASPASE3), counting was performed manually. For markers quantified by fluorescent intensity (DCX, IBA1, GFAP), a region of interest was drawn in Zen software, and the average intensity over that region was recorded. For DCX, the region of interest included the full DGC layer and SGZ. Background autofluorescence was corrected by recording the intensity of a neighboring region of CA3 or hilus devoid of DCX+ cells. For IBA1 and GFAP, the region of interest was the SGZ and hilus, bounded by the inner edge of the granule cell layer and a line drawn between the endpoints of the two blades. No background correction was performed for inflammation markers due to the relatively complete tiling of glia throughout the hippocampus. For each brain, two to five images were quantified per side. A blinded observer quantified all images.

Images for analysis of pyknosis and karyorrhexis were obtained on an Airyscan 880 microscope using a 40× objective. Z-stacks were obtained through the entirety of the DGC layer, tiles were stitched using Zen software, and each individual slice of the z-stack was examined. Nuclei were considered abnormal if the DAPI channel showed condensed, uniform labeling throughout the nucleus instead of the typical variation in intensity observed in healthy cells or if nuclei appeared to be fragmenting into uniformly labeled pieces (*Bayer and Altman, 1974*; *Cahill et al., 2017*). Two blinded observers quantified these images.

Images for analysis of viral tropism were collected on an Airyscan 880 microscope using a 63× objective, and colocalization of cell-type markers with viral GFP was quantified manually by examination of stitched images.

## AAV empty capsid

rAAV8 empty capsids were synthesized and purified using standard CsCl rAAV production protocols by the Salk Viral Core, without the addition of any ITR containing plasmids or sequences.

Electron microscopy quantification was performed at the Salk Institute's Waitt Advanced Biophotonics Center. 3.5 µL of 3% diluted rAAV empty capsid stock or positive control using viral stock of known concentration (AAV1-CAG-flexGFP) was applied to plasma etched carbon film on 200 mesh copper grids (Ted Pella, 01840 F), four grids per stock. Samples were washed three times for 5 s, stained with 1% Uranyl Acetate for 1 min, wicked dry with #1 Whatman filter paper, and air dried before TEM exam. For each grid, four fields were selected in each of four grid squares, and for a total of 16 micrographs per grid, 20,000× magnification on a Libra 120kV PLUS EF/TEM (Carl Zeiss), 2kx2k CCD camera. Two blinded observers each quantified all images, and the ratio of empty to control virus was calculated.

## Cell culture

Mouse NPCs were obtained from E15-E16 embryonic C57BL/6 mouse and cultured as described previously (*Ray and Gage, 2006*), but eliminating the Percoll density gradient centrifugation. NPCs were cultured in DMEM/F-12 supplemented with N2 and B27 (Invitrogen) in the presence of FGF2 (20 ng/mL), EGF (20 ng/mL), laminin (1 µg/mL), and heparin (5 µg/mL), using poly-ornithine/laminin (Sigma)-coated plastic plates. Medium was changed every 2 days, and NPCs were passaged with Accutase (StemCell Tech) when plates reached confluence. Cell cultures underwent regular testing for the presence of Mycoplasma.

## In vitro rAAV transduction and time lapse imaging

NPCs were seeded onto 96-well plates at a density of 10 k cells/well for 24 hr. At 24 hr, medium was changed and supplemented with propidium iodide (1 µg/mL). To serve as baseline, two sets of images were acquired 4 hr apart in bright field and red-fluorescence: five images per well, eight wells per treatment, for five treatments, on an IncuCyte S3 Live Cell Analysis System (Essen Biosciences, Salk Stem Cell Core and UCSD Human Embryonic Stem Cell Core). AAV1-CAG-flex-eGFP stock was serially diluted into sterile PBS (Corning, 21–040-CMR) with an initial MOI at 1 e7, 1 e6, 1 e5, 1 e4. Equal volumes of viral solution or PBS, <1% of the total volume of each well, were added to wells with established cells. MOIs were calculated by dividing total viral particles added per well (1 ε11, 1 ε10, 1 ε9, and 1 ε8 and zero viral particles, respectively) divided by the initial seeding density of 10 k cells/well; the in vitro estimate for MOI inflates the number of viral genomes per cell compared to the in vivo estimate for two reasons. First, cells were allowed to proliferate for 28 hr (approximately a 2–3× increase in cells, *Figure 3B*) before adding virus. Second, in vitro viral particles were distributed throughout the growth media and stochastic diffusion was likely to act as a rate-limiting step to viral entry, whereas in vivo viral adsorption was more likely to act as the rate-limiting step to viral entry. Images were then acquired every 4 hr for 60 hr. Data were extracted using IncuCyte Analysis software.

## In vitro ITR electroporation imaging and FACS analysis

NPCs were collected in equal volumes of nucleofection solution (Amaxa Mouse NSC Nucleofector Kit, Lonza) and electroporated with 5ε6 or 1ε6 copies/cell of 5' biotinylated 145 bp AAV2 ITR ssDNA or scrambled control (ITR: 5'-Biotin-AGGAACCCCTAGTGATGGAGTTGGCCACTC CCTCTC TGCGCGCTCGCTCGCTCACTGAGGCCGGGCGACCAAAGGTCGCCCGACGCCCGGGC TTTGCCCGGGCGGCCTCAGTGAGCGAGCGAGCGCGCAGAGAGGGAGTGGCCAA-3', scramble: 5'-Biotin-CCACATACCGTCTAACGTACGGATTCCGATGCCCAGATAT ATAGTAGATGTCTTATTTG TGGCGGAATAGCGCCAGAGCGTGTAGGCCAACCTTAGTTCTCCATGGAAGGCATCTACCGAAC TCGGTTGCGCGGCCAAATTGGAT-3', Integrated DNA technologies) diluted in sterile PBS or PBS control, in triplicate, then separately plated onto 24-well plates (see schematic in *Figure 3D*). AAV2 ITR sequences were obtained from NCBI Viral Genome database, NC_001401.2 (*Brister et al., 2015*). AAV2 ITRs or a sequence largely homologous with the AAV2 ITR sequence were used in the vast majority of rAAV plasmids. Twenty-four-well plates were segregated for imaging and FACS experiments. Imaging plates were supplemented with propidium iodide and images were acquired on an IncuCyte S3 Live Cell Analysis System as follows: 16 images per well, four wells per replicate,

three replicates per treatment, for five treatments, every 3 hr for 60 hr. Data were extracted using Incucyte Analysis software using the same mask definition obtained above. NPCs from FACS plates were collected in PBS at 12, 24, and 36 hr using Accutase. After incubation for 30 min at RT with Vybrant DyeCycle Green Stain (ThermoFischer, 1:2000), Zombie UV Fixable Viability Kit (BioLegend, 1:1000) and CountBright Absolute Counting Beads (~5000 beads/sample, Thermofisher), cells were filtered into polypropylene FACS collection tubes and FACS analysis was performed on an LSRFortessa X-20 (BD Biosciences, UCSD Human Embryonic Stem Cell Core). Samples were collected by gating on 1000 CountBright Counting Bead counts per well, one well per replicate, three replicates per treatment, for the five treatments, interleaved, at three time points. Populations of live and dead cells (UV Zombie negative and positive cells, respectively), and G1-phase and replicating (S- and G2-phase) cells (Vybrant DyeCycle Green low and high, respectively) were determined using FlowJo software.

## Statistical analysis

All data are presented as mean ± s.d. when describing data between individual samples and as mean ± s.e.m when comparing between groups. To compare histology data across experiments, counts and intensity measures for the injected side of the DG are presented as a percentage of that experiment's mean counts or intensity on the control side. Statistical comparisons were performed in Prism 9.0 (GraphPad Software) using paired t-test (paired data, one independent variable: treatment), repeated measures two-way ANOVA using either the Tukey or Sidak multiple comparison test (two independent variables: treatment and time), or two-way ANOVA using Dunnett's or Tukey's multiple comparison test (in vitro rAAV transduction relative to PBS control and electroporation, respectively) when interaction was significant. Linear regression was performed for BrdU vs cFOS activation. K–S tests were performed for cumulative distributions. All statistical tests were two-tailed. Data was assumed to be normal, and normality tests were not performed. Threshold for significance ($\alpha$) was set at 0.05; * is defined as $p<0.05$, ** is defined as $p<0.01$, *** is defined as $p<0.001$, n.s. is not significant.

## Acknowledgements

We thank Dr. Matt Hirsch, Dr. Jude Samulski, Dr. Tomo Toda, Dr. Simon Schafer, Dr. Carol Marchetto, Lynne Moore, Ruth Keithley for experimental advice; Adrian Martinez for help with histology; and Mary Lynn Gage and Dr. Marianna Alperin for comments on the manuscript. This work was made possible by our funding sources: NIH R01 AG056306, NIH R01 MH114030, NIH K08 NS093130, NIH Shared Instrumentation Grant S10OD025060, The McKnight Endowment Fund for Neuroscience, Whitehall Foundation, JPB Foundation, Annette Merle-Smith, James S McDonnell Foundation, Mathers Foundation, the Leona M and Harry B Helmsley Charitable Trust grant # 2012-PG-MED002, Ray and Dagmar Dolby Family Fund, Salk Innovation Grant, and the Dan and Martina Lewis Biophotonics Fellows Program. Special thanks to Salk Cores: Waitt Advanced Biophotonics Core with funding from NIH-NCI CCSG: P30 014195 and the Waitt Foundation, STEM Core, Viral Vector Core supported by Salk Cancer Center, NCI P30 CA014195; and UCSD Human Embryonic Stem Cell Core.

## Additional information

### Funding

| Funder | Grant reference number | Author |
| --- | --- | --- |
| National Institutes of Health | R01 AG056306 | Fred H Gage |
| National Institutes of Health | R01 MH114030 | Fred H Gage |
| National Institutes of Health | K08 NS093130 | Matthew Shtrahman |
| McKnight Endowment Fund for Neuroscience | | Fred H Gage<br>Matthew Shtrahman |
| National Institutes of Health | S10OD025060 | Matthew Shtrahman |

| | | |
|---|---|---|
| James S. McDonnell Foundation | | Fred H Gage |
| Leona M. and Harry B. Helmsley Charitable Trust | | Fred H Gage |
| Ray and Dagmar Dolby Family Fund | | Fred H Gage |
| Dan and Martina Lewis | Biophotonics Fellows Program | Stephen Johnston |
| Whitehall Foundation Research Grant | 2019-05-71 | J Tiago Gonçalves |

The funders had no role in study design, data collection and interpretation, or the decision to submit the work for publication.

### Author contributions

Stephen Johnston, Conceptualization, Data curation, Software, Formal analysis, Investigation, Visualization, Methodology, Writing - original draft, Writing - review and editing; Sarah L Parylak, Conceptualization, Data curation, Formal analysis, Supervision, Investigation, Visualization, Methodology, Writing - original draft, Project administration, Writing - review and editing; Stacy Kim, Data curation, Formal analysis, Investigation, Visualization, Methodology, Writing - review and editing; Nolan Mac, Christina Lim, Iryna Gallina, Cooper Bloyd, Christian D Saavedra, Investigation; Alexander Newberry, Ondrej Novak, Software; J Tiago Gonçalves, Conceptualization, Investigation, Methodology, Writing - review and editing; Fred H Gage, Conceptualization, Supervision, Funding acquisition, Methodology, Project administration, Writing - review and editing; Matthew Shtrahman, Conceptualization, Data curation, Software, Formal analysis, Investigation, Supervision, Funding acquisition, Methodology, Writing - original draft, Project administration, Writing - review and editing

### Author ORCIDs

Fred H Gage (iD) https://orcid.org/0000-0002-0938-4106
Matthew Shtrahman (iD) https://orcid.org/0000-0003-3185-890X

### Ethics

Animal experimentation: This study was performed in strict accordance with the recommendations in the Guide for the Care and Use of Laboratory Animals of the National Institutes of Health. All of the animals were handled according to approved institutional animal care and use committee (IACUC) protocols (S12201) of the University of California, San Diego. All surgery was performed under isoflurane anesthesia, and every effort was made to minimize suffering.

### Decision letter and Author response

Decision letter https://doi.org/10.7554/eLife.59291.sa1
Author response https://doi.org/10.7554/eLife.59291.sa2

## Additional files

### Supplementary files

• Supplementary file 1. Tukey's test for group comparisons of AAV ITR and SCR control. Post hoc comparisons of the electroporation groups presented in *Figure 3E,F* following two-way ANOVA. *$p<0.05$, **$p<0.01$, ***$p<0.001$, n.s. = not significant.

• Transparent reporting form

### Data availability

All data generated or analysed during this study are included in the manuscript and supporting files. Source data files have been provided for all figures and figure supplements.

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

# Appendix 1

## List of reagents used in the study including antibodies, compounds, viruses, and other key resources

**Appendix 1—key resources table**

| Reagent type (species) or resource | Designation | Source or reference | Identifiers | Additional information |
|---|---|---|---|---|
| Strain, strain background (*Mus musculus*) | C57BL/6J | The Jackson Laboratory | #000664 | |
| Strain, strain background (*Mus musculus*) | STING-KO (B6(Cg)-*Sting1$^{tm1.2Camb}$*/J) | The Jackson Laboratory *Jin et al., 2011* | #025805 | |
| Chemical compound, drug | BrdU (5-Bromo-2'-deoxyuridine) | Millipore Sigma | B5002 | 50 mg/kg/day i.p. for 3 days |
| Other | Nanoject III | Drummond Science | 3-000-207 | |
| Strain, strain background (*AAV*) | AAV1-CAG-GFP | Addgene | 37825 | |
| Strain, strain background (*AAV*) | AAV1-CAG::flex-eGFP-WPRE-bGH | Addgene and University of Pennsylvania | 51502 | |
| Strain, strain background (*AAV*) | AAVretro-CaMKIIa::NES-jRGECO1a-WPRE-SV40 | Gage, Salk Viral Core | | |
| Strain, strain background (*AAV*) | AAV8-CaMKIIa::NES-jRGECO1a-WPRE-SV40 | Gage, Salk Viral Core | | |
| Strain, strain background (*AAV*) | AAV1-Syn::NES-jRGECO1a-WPRE-SV40 | Addgene and University of Pennsylvania | 100854 | |
| Strain, strain background (*AAV*) | AAV8-CaMKIIa::mCherry-WPRE-bGH | Addgene and Salk Viral Core | 114469 | |
| Strain, strain background (*AAV*) | AAV8 capsid | University of North Carolina Viral Vector Core pXR8, Salk Viral Core | | |
| Other | 3 mm glass coverslip | Warner Instruments | CS-3R | #1 thickness |
| Other | Optibond All-In-One | Kerr Dental | 33381 | |
| Other | 2-Photon Laser Scanning Microscope Movable Objective Microscope (MOM) | Sutter Instruments | | |
| Other | 1070 nm femtosecond-pulsed laser | Coherent | Fidelity-2 | |
| Other | Nikon 16X water immersion objective | 0.8 NA, Nikon | CFI75 LWD 16X W | |
| Software, algorithm | Matlab | MathWorks | | |

*Continued on next page*

*Appendix 1—key resources table continued*

| Reagent type (species) or resource | Designation | Source or reference | Identifiers | Additional information |
|---|---|---|---|---|
| Software, algorithm | FlowJo | BD Biosciences | | |
| Software, algorithm | Prism 9.0 | GraphPad | | |
| Antibody | Anti-BrdU (rat monoclonal) | Accurate. Novus, Abcam | OBT0030, Accurate; NB500-169, Novus; AB6326, Abcam | IHC (1:250) |
| Antibody | Anti-cleaved-CASPASE3 (rabbit polyclonal) | Cell Signaling | 9661 | IHC (1:200) |
| Antibody | Anti-CFOS (goat polyclonal) | Santa Cruz | sc-52-G | IHC (1:500) |
| Antibody | Anti-CFOS (rabbit polyclonal) | Synaptic Systems | 226003 | IHC (1:500) |
| Antibody | Anti-DCX (goat polyclonal) | Santa Cruz | sc-8066 | IHC (1:50–500) |
| Antibody | Anti-DCX (guinea pig polyclonal) | Millipore | AB2253 | IHC (1:500) |
| Antibody | Anti-GFAP (chicken polyclonal) | Millipore | AB5541 | Ihc (1:1000) |
| Antibody | Anti-GFP (chicken polyclonal) | Aves Labs | GFP-1020 | IHC (1:1000) |
| Antibody | Anti-PROX1 (rabbit polyclonal) | Abcam | ab101851 | IHC (1:500) |
| Antibody | Anti-SOX2 (rabbit polyclonal) | Cell Signaling | 2748 | IHC (1:250) |
| Antibody | Anti-TBR2 (rabbit monoclonal) | Abcam | ab183991 | IHC (1:500–1000) |
| Antibody | Anti-SOX2 (rat monoclonal) | Invitrogen | 14981182 | IHC (1:1000) |
| Antibody | Anti-chicken-AlexaFluor488 (donkey polyclonal) | Jackson Immuno Research Laboratories | 703-545-155 | IHC (1:250) |
| Antibody | Anti-rat-AlexaFluor647 (donkey polyclonal) | Jackson Immuno Research Laboratories | 712-605-153 | IHC (1:250) |
| Antibody | Anti-rabbit-Cy5 (donkey polyclonal) | Jackson Immuno Research Laboratories | 711-175-152 | IHC (1:250) |
| Antibody | Anti-rabbit-Cy3 (donkey polyclonal) | Jackson Immuno Research Laboratories | 711-165-152 | Ihc (1:250) |
| Antibody | Anti-rabbit-AlexaFluor488 (donkey polyclonal) | Jackson Immuno Research Laboratories | 711-545-152 | IHC (1:250) |
| Antibody | Anti-guinea pig-AlexaFluor488 (donkey polyclonal) | Jackson Immuno Research Laboratories | 706-545-148 | IHC (1:250) |
| Antibody | Anti-guinea pig-Cy3 (donkey polyclonal) | Jackson Immuno Research Laboratories | 706-165-148 | IHC (1:250) |
| Antibody | Anti-guinea pig-AlexaFluor647 (donkey polyclonal) | Jackson Immuno Research Laboratories | 706-605-148 | IHC (1:250) |
| Antibody | Anti-goat-Cy5 (donkey polyclonal) | Jackson Immuno Research Laboratories | 705-175-147 | IHC (1:250) |

*Appendix 1—key resources table continued*

| Reagent type (species) or resource | Designation | Source or reference | Identifiers | Additional information |
|---|---|---|---|---|
| Antibody | anti-goat-Cy3 (donkey polyclonal) | Jackson Immuno Research Laboratories | 705-165-147 | IHC (1:250) |
| Antibody | Anti-goat-AlexaFlour488 (donkey polyclonal) | Jackson Immuno Research Laboratories | 705-545-147 | IHC (1:250) |
| Other | Zeiss laser scanning confocal microscope | Zeiss | LSM 710, LSM 780, or Airyscan 880 | 20x and 63x objective |
| Other | Slide scanning microscope | Olympus | VS-120 | 10× objective |
| Other | Libra 120kV PLUS EF/TEM | Carl Zeiss | | 2kx2k CCD camera, 20,000x magnification |
| Commercial assay or kit | Amaxa Mouse NSC Nucleofector Kit | Lonza | VPG-1004 | |
| Sequenced-based reagent | ITR | Integrated DNA Technologies | 20nmole Ultramer DNA Oligo | 5'-Biotin- AGGAACCCCTAGTGATGGAGTTGGCCACTCCCTCTCTGCGCGCTCGCTCGCTCACTGAGGCCGGGCGACCAAAGGTCGCCCGACGCCCGGGCTTTGCCCGGGCGGCCTCAGTGAGCGAGCGAGCGCGCAGAGAGGGAGTGGCCAA-3' |
| Sequenced-based reagent | ITR Scramble | Integrated DNA Technologies | 20nmole Ultramer DNA Oligo | 5'-Biotin-CCACATACCGT\CTAACGTACGGATTCCGATGCCCAGATATATAGTAGATGTCTTATTTGTGGCGGAATAGCGCCAGAGCGTGTAGGCCAACCTTAGTTCTCCATGGAAGGCATCTACCGAACTCGGTTGCGCGGCCAAATTGGAT-3' |
| Chemical compound, drug | Vybrant DyeCycle Green Stain | Thermo Fisher | V35004 | 1:2000 |
| Chemical compound, drug | Zombie UV Fixable Viability Kit | BioLegend | 423107 | 1:1000 |
| Other | CountBright Absolute Counting Beads | Thermo Fisher | C36950 | ~5000 beads/sample |
| Other | LSRFortessa X-20 | BD Biosciences and UCSD Human Embryonic Stem Cell Core | | |

