## [Decision Letter]

**Acceptance summary:**

Johnston et al. focused on adeno-associated virus (AAV) that is commonly used to study adult neurogenesis in the rodent hippocampus. They show that the neural progenitor cells and immature dentate granule cells (DGCs) present in the hippocampal dentate gyrus (DG) subregion, are particularly sensitive to rAAV-induced cell death. In response to the loss of these populations, the remaining, mature DGCs became hyperactive for some time. To circumvent these effects and still transduce the DG efficiently, the authors show that injection of rAAV2-retro serotyped virus into the nearby CA3 subregion of the hippocampus, does result in an efficient retrograde labeling of the mature DGCs, while keeping adult neurogenesis intact. The paper reports on important effects of a commonly used approach in neuroscience and stem cell biology. Next to highlighting this effect of AAV, also their demonstration of an effective, alternative route of transduction of the DG will be highly valuable for the field.

**Decision letter after peer review:**

Thank you for submitting your article "AAV Ablates Neurogenesis in the Adult Murine Hippocampus" for consideration by *eLife*. Your article has been reviewed by 3 peer reviewers, one of whom is a member of our Board of Reviewing Editors, and the evaluation has been overseen by Laura Colgin as the Senior Editor. The reviewers have opted to remain anonymous.

The reviewers have discussed the reviews with one another and the Reviewing Editor has drafted this decision to help you prepare a revised submission.

As the editors have judged that your manuscript is of interest, but as described below, additional experiments are required before it can be published.

We would like to draw your attention to changes in our revision policy that we have made in response to COVID-19 (https://elifesciences.org/articles/57162). First, because many researchers have temporarily lost access to the labs, we will give authors as much time as they need to submit revised manuscripts. We are also offering, if you choose, to post the manuscript to bioRxiv (if it is not already there) along with this decision letter and a formal designation that the manuscript is "in revision at eLife". Please let us know if you would like to pursue this option. (If your work is more suitable for medRxiv, you will need to post the preprint yourself, as the mechanisms for us to do so are still in development.)

Summary:

Johnston et al. demonstrate that hippocampal neural progenitor cells and immature dentate granule cells are particularly sensitive to rAAV-induced cell death at experimentally relevant titers. Also in vitro application of AAV or electroporation of AAV2 inverted terminal repeats (ITRs) is sufficient to induce cell death. The remaining mature DGCs appear hyperactive for a prolonged period of time. To circumvent these effects and transduce the dentate gyrus (DG) while keeping adult neurogenesis intact, injection of rAAV2-retro serotyped virus into CA3 was shown to result in efficient retrograde labeling of mature DGCs.

This paper reports on an important tool for neuroscience and stem cell biology in general, and studies AAV effects in a multidisciplinary manner with extensive experimental detail; the approaches selected all provide relevant information, most (contralateral, etc) controls are done and they have tested an extensive time series and different concentrations; the data is solid, convincing and relevant and as such, warrants publication in *eLife*.

Essential Revisions:*Reviewer #1:*

1) One minor issue not discussed much now is their use of isoflurane and dexamethasone during surgery (p.25). Both compounds are well known to alter/suppress neurogenesis per se and while the authors show the virus has not introduced lasting inflammatory responses (based on late microglia markers), and while contralateral controls were done, viral application has occurred in the context of stress and selective GR activation, local insult and immunosuppression, which could at least possibly, have contributed to (changes in) the properties (Fitzsimons MP 2013; Lucassen CSHPB 2015) and selective vulnerability of the NPC and DGC populations, or their local environment, i.e. the adult DGCs. As they do compare 'positive' stimuli like exercise and enrichment, possible effects of suppressive stimuli like stress and immune suppression, deserve to be discussed as well.

2) The authors do not show now whether AAV has actually targeted the different cell types in vivo directly, or whether the current effects may possibly arise from indirect, local effects.

3) In this respect, also the prolonged hyperactivity of the remaining DG cells indicates that at least the local surroundings of the NPCs have been altered, which could potentially affect chances of survival for new cells born in that context. This, and the reasons for the remaining hyperactivity, could be discussed some more; several hypotheses have been proposed in literature.

4) Are differences between dorsal and ventral hippocampus involved?*Reviewer #2:*

In vivo studies. This paper describes a strong and unexpected decline in neural progenitor cells (NPCs) and immature dentate granule cells in adult mouse hippocampus following injection of adeno-associated viral vectors (AAVs) in the dentate gyrus. The decline in these cells is very rapid (within 18 hours post-injection of AAV) and persistent. Mature dentate gyrus neurons express c-fos at 4 weeks post-injection of AAV, a sign that they are hyperactive. An early decline in Tbr2 cells (within 18 – 24 hours after AAV delivery) is followed 4 weeks later by an inflammatory response. Transduction of adult dentate granule cells with AAV2retro has no deleterious effects. These observations are an indication that AAV-mediated transduction of cells in the dentate gyrus interferes with neurogenesis and results in changes in hippocampal circuitry, whereas selective transduction of only adult dentate granule cells via the retrograde route has no negative effect on neurons in the dentate gyrus.

What is lacking in these in vivo studies is evidence that the AAV serotypes used are indeed transducing the early (*Sox2*), middle (Tbr2) and late (DCX+) stage cells in the dentate gyrus. This is important for the interpretation of the data. The number of *sox2*^+^ cells was hardy reduced following AAV injection and this is taken as evidence that these cells are resistant to AAV toxicity. However an alternative explanation could be that these cells are not or poorly transduced by AAV1 or AAV8. Most Tbr2 cells disappear acutely within 48 hours after AAV delivery, whereas DCX cells decline gradually. This is interpreted as a failure of the Tbr2 cells to replenish the DCX cell population (discussion page 19, lines 5 to 17). An alternative explanation could be that this difference is caused by a differential sensitivity to AAV or differential transduction efficiencies of these two populations of immature neurons by AAV.

The early decline in Tbr2 cells is followed 4 weeks later by an inflammatory response. Is this inflammatory response the result of the death of immature neurons or is this a response to AAV? Could the inflammation in the dentate gyrus create a non-permissive environment for neurogenesis and could this be the reason for the failure of the *Sox2* cells to replenish the Tbr2 and DCX cell populations? These issues should be discussed in the discussion.

in vitro studies. In search for the mechanism underlying AAV-mediated death of NPCs, a separate set of in vitro studies were performed that show that transfection of AAV-ITRs is sufficient to induce cell death in cultured embryonic hippocampal mouse NPCs. This observation is in line with a previous report by Hirsch et al. (2011) showing that a 40 nucleotide G-quadruplex rich telomere-like sequence in the AAV-ITR triggers an apoptotic response specifically in pluripotent human embryonic stem cells. It remains unclear, however, whether ITR-induced apoptosis is responsible for the selective death of Tbr2+ cells in the adult hippocampus as observed in the in vivo studies.*Reviewer #3:*

The findings are very interesting and provide further evidence for the risks associated with the use of rAAV vectors, especially for human gene therapy, and for this reason, are relevant for a broad audience. The manuscript is well written and although complex, easy to read and follow. however, there are a number of points that require the authors' attention.

Some of the data presented in the initial figures of the paper require improvement as they at points, contradict later figures:

1. At the beginning of the Results section, the authors state that transduction with rAAVs resulted in ablation of adult neurogenesis. This seems like a conclusion, as the images presented in Sup figure 1 are DCX+ cells and these are not quantified. It would be more careful to refer to DCX+ numbers rather than to adult neurogenesis. In the same line, the authors state that the effect they observe was robust regardless of production facility, purification method, promoter used, etc…This could be rephrased as these factors were not independent of each other, at least as the data is presented in Sup Figure 1 (three production methods tested at Salk, unknown production method at U Penn, AAV8 used at Salk, AAv1 at U Penn). Finally, the authors mention they have also tested AAV9, but the data is not shown. All these points make the Sup Figure 1 feels somewhat disorganized and confusing. What do the Bule and red symbols in Sup Figure 1B represent?

2. In figure 1 the authors concentrate on the effect of transduction with rAAVs on the survival of cells previously labeled by BrdU. However, I could not find any indication that BrdU^+^ cells were actually transduced with the rAAVs in Figure 1 B/D. Are BrdU^+^ cells positive for GFP? To which percentage? This should be quantified to support the later claims of cell-intrinsic effects.

3. The effects of transduction with rAAV on cell survival were all quantified in BrdU^+^ cells. As some toxicity of BrdU administration on neural precursors has been reported before (e.g. Caldwell et al., 2005; Kolb JNS99), it can not be discarded that the reported effects may be, at least in part, the results of a combined toxicity induced by the combination of BrdU and transduction with rAAV. To discard this possibility the authors could use a different birthdating/labeling method, preferably genetic, to avoid additional toxicity. This point seems particularly important as the main effect on cell loss is on proliferating Tbr2+ cells ("proliferating cells are the main target of the virus", page 9).

4. On page 10 the authors state that there was no effect on the expression of the microglia marker Iba1 until 4 weeks p.i. This seems contradictory with the data presented in Sup Figure 1 in which the authors show that transduction with AAV8-CaMKiia-NES-jRGECO (Salk Viral Core – CsCl purification) demonstrated variability in magnitude but consistency in direction of Iba1+ microglia activation. Could the authors please clarify this apparent contradiction?

5. On page 10, the authors describe no obvious change in microglia morphology 2 days or 1 week p.i. Where is this shown and how was this quantified? Similarly, on page 11 the authors state that at 4 weeks microglia exhibited an ameboid morphology. Where is this shown?

6. The water (H20) control used in the in vitro experiments presented in figure 3 seems questionable. Usually, in this kind of experiments virus are resuspended in culture medium to avoid changes in Ph, salinity, etc in the cell culture. Which was the volume used to dissolve the viral suspensions? Could these additions have had consequences on the composition of the cell culture medium? Please clarify this technical point.

7. The authors show that the rAAVs may be toxic for NPSc and immature neurons derived from them, but there is no information presented on the effect of transduction with rAAvs on mature dentate granule neurons, neither of embryonic origin nor originated from adult neurogenesis. The authors should present data on the effects on these relevant cell types.

8. Is transduction with rAAV also toxic for CA3 neurons? This is an important point regarding the concerns the authors' data raise on the general applicability of AAVS fro human gene therapy.

9. The hypothesis that infected Type 1 cells undergo apoptosis, precluding neurogenesis, is testable.

---

## [Author Response]

Reviewer #1:1) One minor issue not discussed much now is their use of isoflurane and dexamethasone during surgery.

Previously, we examined the effect of similar surgeries, including the use of isoflurane and dexamethasone, on activity-dependent development of adult-born dentate granule cell (DGC) dendritic arbors, which showed no discernible effects from anesthesia or surgical procedure conditions (Goncalves et al., Nat. Neuroscience 2016, Supplementary Figure 8,9). However, we agree that there is relevant literature indicating that there could be “selective vulnerability of the NPC and DGC populations” to these experimental manipulations. On page 2 and page 13 of the revised manuscript, we discuss the “possible effects of suppressive stimuli like stress and immune suppression.”

These stem cells and their immature progeny are sensitive to environmental stimuli; their proliferation, development, and survival are regulated by multiple intrinsic and extrinsic factors, including experience, stress, inflammation, and pharmacologic agents.

In any surgical manipulation of the DG, including our studies, abDGCs are exposed to a variety of experimental manipulations that might affect adult neurogenesis, including anesthesia, nonsteroidal anti-inflammatory drugs, and corticosteroids utilized for animal comfort and humane experimentation (Cameron and Gould, 1994; Erasso et al., 2013; Kim et al., 2020; Lehmann et al., 2013; McGuiness et al., 2017; Monje et al., 2003; Saaltink and Vreugdenhil, 2014; Schoenfeld and Gould, 2013; Stratmann et al., 2009, 2010). The effects of these pharmacological agents are largely accounted for by performing intra-subject comparisons where the uninjected or saline-injected contralateral hippocampus is also exposed to these agents. Previous work indicates that these compounds have no significant effect on the development of dendritic arbors in abDGCs, which are sensitive to experience (Gonçalves et al., 2016a).

2) The authors do not show now whether AAV has actually targeted the different cell types in vivo directly, or whether the current effects may possibly arise from indirect, local effects.

The concerns regarding the lack of direct demonstration of viral transduction and the possible role of differences in viral tropism across the different cell types described in the study were shared among the reviewers. As described in the manuscript, the cell populations most vulnerable to AAV infection, including recently labeled BrdU+ cells and Tbr2 + cells, are eliminated within 48 hours of infection. In fact, our unpublished data indicate that these populations are almost entirely absent at 24 hours post-infection. This timeframe does not provide enough time for sufficient expression of AAV-delivered transgenes and, as a result, there is no straightforward method for accurate demonstration and quantification of infection in vulnerable cell populations prior to elimination.

Despite this limitation, we were able to address the question of whether some cell types might avoid AAV toxicity through a “decreased sensitivity to AAV or differential transduction” (Reviewer #2). We performed experiments delivering AAV expressing GFP into the dentate gyrus and measured the percentage of sox2+, DCX+, and Tbr2+ cells in the subgranular zone that express GFP 1 week after viral injection. These experiments demonstrate that the majority of sox2+ cells remaining 1 week after rAAV infection (the minimal time required for reliable GFP protein expression) are GFP positive and that these cells survive despite being infected by rAAV. Overall, we observed an increasing percentage of GFP-expressing cells with increasing survival across the three populations, suggesting that differences in viral tropism are not likely to explain the measurable differences in AAV toxicity (page 5, Figure 2—figure supplement 1F). However, due to strong survivor bias, particularly with Tbr2+ and DCX+ cells, we are not able to accurately quantify transduction efficiency across cell-types.

We addressed the question of “indirect, local effects” in Supplementary Figure 2I of the original manuscript (now Figure 2—figure supplement 2E), where 30 nanoliter AAV injections were performed in the dentate gyrus. These experiments demonstrate loss of “DCX cells within the area of viral spread” and sparing of “uninfected cells located microns away from those expressing jRGECO1a”. We have edited the description slightly on page 7 to emphasize indirect and local effects:

“These findings suggest that AAV-induced toxicity may be cell autonomous and is unlikely to be mediated by astrocyte- or microglia-activated immune responses or by inflammatory signals and other indirect or local changes within the niche”

In addition, dose-dependent AAV-induced toxicity was observed in vitro, again suggesting that indirect and local effects arising from the niche or cells other than NPCs are unlikely to mediate AAV-induced toxicity.

3) In this respect, also the prolonged hyperactivity of the remaining DG cells indicates that at least the local surroundings of the NPCs have been altered, which could potentially affect chances of survival for new cells born in that context. This, and the reasons for the remaining hyperactivity, could be discussed some more; several hypotheses have been proposed in literature.

As we demonstrated in the manuscript, neurogenesis is essentially absent 4 weeks after “high dose” (3E12 gc/ml) AAV injection, when hyperactivity is observed in mature DGCs. While we agree that hyperactivity resulting 4 weeks after partial ablation of neurogenesis using intermediate doses of AAV (1 E12 gc/ml) could potentially enhance the survival of DGCs born around or slightly before the period of hyperactivity, there is a significant amount of inflammation 4 weeks after injection that would attenuate neurogenesis, even at intermediate doses. Thus, there is likely to be a complex combination of effects and we cannot comment with any confidence about the “chances of survival for new cells born in that context”. With regards to reasons for DG hyperactivity, we have updated the discussion (page 11) to read:

“Moreover, DG activity and computations appear to depend on the addition of abDGCs (Clelland et al., 2009; Ikrar et al., 2013; Sahay et al., 2011), whose net effect is to quiet activity in mature DGCs and the rest of the hippocampus (Berdugo-Vega et al., 2020). This inhibition on mature DGCs through either monosynaptic (Luna et al., 2019) or polysynaptic inhibition (Jinde et al., 2013; Toni et al., 2008) is thought to enhance pattern separation by selectively suppressing competing engrams (Espinoza et al., 2018; Johnston et al., 2016; McAvoy et al., 2016; Sahay et al., 2011). Consistent with this idea, rAAV-induced ablation of neurogenesis results in a dose-dependent increase in mature DGC activity 4 weeks after infection, as measured by immediate early gene expression (Figure 4A,B, Figure 4—figure supplement 1A).”

4) Are differences between dorsal and ventral hippocampus involved?

Viral transduction was detected throughout the dorsal dentate gyrus and typically spread to the intermediate dentate gyrus (see Figure 1—figure supplement 1E), a region that some delineate as ventral dentate gyrus (Piatti et al., J. Neurosci 2011, Hawley and Leasure, Hippocampus 2011). We saw no obvious differences in rAAV-induced toxicity along the extent of the viral spread.

Reviewer #2:In vivo *studies. This paper describes a strong and unexpected decline in neural progenitor cells (NPCs) and immature dentate granule cells in adult mouse hippocampus following injection of adeno-associated viral vectors (AAVs) in the dentate gyrus. The decline in these cells is very rapid (within 18 hours post-injection of AAV) and persistent. Mature dentate gyrus neurons express c-fos at 4 weeks post-injection of AAV, a sign that they are hyperactive. An early decline in Tbr2 cells (within 18 – 24 hours after AAV delivery) is followed 4 weeks later by an inflammatory response. Transduction of adult dentate granule cells with AAV2retro has no deleterious effects. These observations are an indication that AAV-mediated transduction of cells in the dentate gyrus interferes with neurogenesis and results in changes in hippocampal circuitry, whereas selective transduction of only adult dentate granule cells via the retrograde route has no negative effect on neurons in the dentate gyrus.*What is lacking in these in vivo studies is evidence that the AAV serotypes used are indeed transducing the early (Sox2), middle (Tbr2) and late (DCX+) stage cells in the dentate gyrus.

Please see response to Reviewer #1, concern #2 above.

This is important for the interpretation of the data. The number of Sox2^+^ cells was hardy reduced following AAV injection and this is taken as evidence that these cells are resistant to AAV toxicity. However an alternative explanation could be that these cells are not or poorly transduced by AAV1 or AAV8. Most Tbr2 cells disappear acutely within 48 hours after AAV delivery, whereas DCX cells decline gradually. This is interpreted as a failure of the Tbr2 cells to replenish the DCX cell population.

Cell-mediated inflammatory responses and toxicity due to recombinant AAV transduction have been described previously in other brain regions devoid of neurogenesis (Hadaczek et al. Hum Gene Therapy. 2009 , Cieseilska et al. Mol Ther. 2013), indicating that the ablation of neurogenesis is not required for inducing subsequent inflammation. However, it could be a contributing factor that enhances the inflammation observed in this brain region. We certainly agree that inflammation could create a “non-permissive environment for neurogenesis and could… be the reason for the failure of the Sox2 cells to replenish the Tbr2 and DCX cell populations”. On page 10 of the discussion, we include the statement:

“Alternatively, delayed inflammation observed in our experiments 1 month after rAAV injection could create a non-permissive environment for neurogenesis and could explain the inability of Sox2 cells to replenish the Tbr2 and DCX cell populations.”

It remains unclear, however, whether ITR-induced apoptosis is responsible for the selective death of Tbr2+ cells in the adult hippocampus as observed in the in vivo studies.

We are eagerly and actively exploring approaches to definitively answer this question in the laboratory. However, it remains a significant technical challenge to deliver oligomers greater than 30 base pairs in length reliably to neurons in vivo. Such experiments are required to demonstrate sufficiency in vivo. However, experiments transducing with empty capsids do demonstrate that ITRs are indeed necessary for ablation of Tbr2+ cells in vivo (Figure 2F). There are a number of proprietary lipid carriers that have been developed for mRNA vaccines, including COVID-19, that we would like to test as non-viral vectors to deliver ITRs in the rodent brain. Currently, these experiments are technically challenging and are likely a few years away. We have revised the manuscript to highlight this limitation in our study (page 10).

Reviewer #3:The findings are very interesting and provide further evidence for the risks associated with the use of rAAV vectors, especially for human gene therapy, and for this reason, are relevant for a broad audience. The manuscript is well written and although complex, easy to read and follow. however, there are a number of points that require the authors' attention.Some of the data presented in the initial figures of the paper require improvement as they at points, contradict later figures:1. At the beginning of the Results section, the authors state that transduction with rAAVs resulted in ablation of adult neurogenesis. This seems like a conclusion, as the images presented in Sup figure 1 are DCX+ cells and these are not quantified. It would be more careful to refer to DCX+ numbers rather than to adult neurogenesis. In the same line, the authors state that the effect they observe was robust regardless of production facility, purification method, promoter used, etc…This could be rephrased as these factors were not independent of each other, at least as the data is presented in Sup Figure 1 (three production methods tested at Salk, unknown production method at U Penn, AAV8 used at Salk, AAv1 at U Penn). Finally, the authors mention they have also tested AAV9, but the data is not shown. All these points make the Sup Figure 1 feels somewhat disorganized and confusing.

We have simplified and revised Figure 1—figure supplement 1 based on the feedback from Reviewer #3.

2. In figure 1 the authors concentrate on the effect of transduction with rAAVs on the survival of cells previously labeled by BrdU. However, I could not find any indication that BrdU^+^ cells were actually transduced with the rAAVs in Figure 1 B/D. Are BrdU^+^ cells positive for GFP? To which percentage? This should be quantified to support the later claims of cell-intrinsic effects.

Please see response to Reviewer #1, concern #2 above.

3. The effects of transduction with rAAV on cell survival were all quantified in BrdU^+^ cells. As some toxicity of BrdU administration on neural precursors has been reported before (e.g. Caldwell et al., 2005; Kolb JNS99), it can not be discarded that the reported effects may be, at least in part, the results of a combined toxicity induced by the combination of BrdU and transduction with rAAV.

We address this concern with new experiments described on page 5 of the revised manuscript, which show significant loss of Tbr2+ and DCX+ cells in the absence of BrdU administration.

Significant loss of Tbr2+ and DCX+ cells occurred even in the most conservative experimental conditions, in which BrdU was absent (to prevent any synergistic toxicity between rAAV and BrdU) and when saline was injected contralaterally (to mimic any physical disruption due to the injection process itself) (Figure 2—figure supplement 1A-E).

4. On page 10 the authors state that there was no effect on the expression of the microglia marker Iba1 until 4 weeks p.i. This seems contradictory with the data presented in Sup Figure 1 in which the authors show that transduction with AAV8-CaMKiia-NES-jRGECO (Salk Viral Core – CsCl purification) demonstrated variability in magnitude but consistency in direction of Iba1+ microglia activation.

Indeed, our early experiments demonstrated that the inflammatory response was highly variable at 2 weeks, with some mice showing clear signs of microglia proliferation and others showing no increase in iba1 staining (see 4th example in Figure 1—figure supplement 1A from Penn Vector core). However, we did not perform systematic experiments at this time point. We have clarified this variability observed at the 2-week time point in the manuscript (page 6).

“Variable inflammatory response was observed with different viral preparations, particularly around the intermediate time point of 2 weeks (Figure 1—figure supplement 1A.). However, rapid (< 48 hours) loss of NPCs (Figure 2) occurred independent of expression of the microglial marker Iba1, which did not increase until 4 weeks post-injection”

5. On page 10, the authors describe no obvious change in microglia morphology 2 days or 1 week p.i. Where is this shown and how was this quantified? Similarly, on page 11 the authors state that at 4 weeks microglia exhibited an ameboid morphology. Where is this shown?

Example images demonstrating changes in microglia morphology are now included in Figure 2—figure supplement 2A.

6. The water (H20) control used in the in vitro experiments presented in figure 3 seems questionable. Usually, in this kind of experiments virus are resuspended in culture medium to avoid changes in Ph, salinity, etc in the cell culture. Which was the volume used to dissolve the viral suspensions? Could these additions have had consequences on the composition of the cell culture medium?

The control condition was listed as water (H20) in error and was indeed sterile phosphate buffer saline (PBS). The manuscript has been corrected to reflect this.

7. The authors show that the rAAVs may be toxic for NPSc and immature neurons derived from them, but there is no information presented on the effect of transduction with rAAvs on mature dentate granule neurons.

Figure 1B demonstrates that cells labeled with BrdU and born 8 weeks prior to AAV injection, which is widely recognized as a time point of full maturity for adult-born dentate granule cells (Laplagne et al., PloS Biol. 2006, Piatti et al., Front Neurosci. 2013), exhibit no discernible toxicity 1 week post infection. This approach provides perhaps the most rigorous method to unambiguously identify and label mature dentate granule cells with a definitive age.

9. The hypothesis that infected Type 1 cells undergo apoptosis, precluding neurogenesis, is testable.

To address this concern, we stained hippocampal slices from mice infected with AAV expressing GFP and sacrificed 1 week later (see response to Reviewer #1 concern #2) for Sox2 and caspase3 to identify sox2+GFP+caspase3+ cells. As described in the manuscript, loss of Sox2 cells was not detectable at 1 week (Figure 2—figure supplement 1C), and triple labeled sox2+GFP+caspase3+ were exceedingly rare. Thus, while we cannot rule out the hypothesis, there is no evidence of Type 1 cells undergoing apoptosis at this time point (described on page 10 of revised manuscript).

“However, few sox2+ cells were lost over the course of 4 weeks (Figure 2c and Supplementary Figure 2c) and triple labeled sox2+GFP+caspase3+ cells 1 week after viral injection were exceedingly rare.”